# Global coastal attenuation of wind-waves observed with radar altimetry

Marcello Passaro [1✉], Mark A. Hemer[2], Graham D. Quartly [3], Christian Schwatke [1], Denise Dettmering [1] & Florian Seitz [1]

Coastal studies of wave climate and evaluations of wave energy resources are mainly regional and based on the use of computationally very expensive models or a network of in-situ data. Considering the significant wave height, satellite radar altimetry provides an established global and relatively long-term source, whose coastal data are nevertheless typically flagged as unreliable within 30 km of the coast. This study exploits the reprocessing of the radar altimetry signals with a dedicated fitting algorithm to retrieve several years of significant wave height records in the coastal zone. We show significant variations in annual cycle amplitudes and mean state in the last 30 km from the coastline compared to offshore, in areas that were up to now not observable with standard radar altimetry. Consequently, a decrease in the average wave energy flux is observed. Globally, we found that the mean significant wave height at 3 km off the coast is on average 22% smaller than offshore, the amplitude of the annual cycle is reduced on average by 14% and the mean energy flux loses 38% of its offshore value.

---

[1] Deutsches Geodätisches Forschungsinstitut der Technischen Universität München, München, Germany. [2] Commonwealth Scientific and Industrial Research Organisation Oceans and Atmosphere, Hobart, Tasmania, Australia. [3] Plymouth Marine Laboratory (PML), Plymouth, UK. ✉email: marcello.passaro@tum.de

The height of the wind waves in the ocean, together with their period, direction, and speed, is a fundamental parameter to describe the sea state and more generally to study the ocean climate and interactions with the atmosphere[1]. The significant wave height (SWH), defined as four times the standard deviation (std) of the surface elevation[2], is an integral parameter that is extensively used as reference to quantify both extremes and mean sea states. In particular, the relevance of a SWH climatology is manifold, from the optimisation of shipping routes[3] to the quantification of the impact of sea level rise at the coast[4,5]. Such a climatology is also fundamental to assess the wave energy resources of a particular area and planning the exploitation of a potential source of renewable energy[6].

Despite the overwhelming importance of measuring waves in the coastal zone, our knowledge of coastal wave climate and coastal wave energy resources is limited in data availability, accuracy, and resolution[2]. In-situ data are collected by buoys, whose records are sparse in time and space. Ocean models work very well in the open ocean, but a nested high resolution model needs to be used in order to correctly simulate the coastal features that modify wave parameters.

Satellite altimeter measurements of SWH, collected along repeating ground tracks, extend from 1985 through to present[7]. The principle is based on the shape of the returned radar echo and in particular on its stretch being proportional to the wave height[8]. Such estimation has the advantage of being independent of atmospheric corrections that are needed to estimate the range (distance between the satellite centre of mass and the sea level). This technique has been used to quantify global open ocean mean wave climate, seasonality[9,10], energy flux resources[11], and global historical trends[12]. Nevertheless, these studies cannot see small scale variability of coastal processes, given the large grid-points of over 1° spacing in latitude and longitude. Other studies identified the potential of using along-track measurements to locally observe variations in the sea state[13,14], but efforts have been restricted to specific regions and were limited by the unreliability of standard altimetry data in the coastal strip. This is due to the influence of land and areas with different backscattering characteristics within the satellite footprint[15], which can negatively affect SWH measurements within about 20 km of the coast[16].

In recent years, coastal altimetry has been the focus of several improvements[17]. In particular, specific algorithms (retrackers) have been designed to fit the returned echo while avoiding spurious coastal reflections that degrade the quality of the estimated parameters. This, coupled with a conservative strategy to detect outliers, has brought strong improvements to the quality and the quantity of SWH retrievals.

Here, we exploit these improvements to provide, based on reprocessed along-track satellite altimetry data, an assessment of the average global coastal wave climate in the coastal zone in terms of SWH, and to highlight the differences with respect to the climatology of previously presented offshore conditions[9,10]. The results presented are based on the reprocessing of satellite altimetry data from Jason-1 and Jason-2 missions, from July 2001 to January 2016, following the methodology described in Section Methods. We are able to quantify the progressive attenuation of the average wave climate towards the coast, even focusing on the differences in the last 30 km. These differences are finally quantified in terms of wave energy flux. The coastal proximity and resolution, as well as the global character of these observations is unprecedented and verified by comparison with buoys and a regional high-resolution nested wave model.

## Results

**Mean significant wave height.** The terminology referring to coastal oceanography as compared to the variability further away from the coast differs significantly in the literature. In this study, we define coastal along-track locations and compare their variability against offshore along-track locations. Coastal wave measurement points are taken as the location of the 1-Hz sample point nearest to the coast (noting points within 3 km of the coast are excluded to avoid outliers). Offshore wave measurement points are taken as the first 1-Hz sample point located more than 30 km from the coast. In order to ease comparison between offshore and coastal points, we consider only along-track sections with a single ocean-land or land-ocean transition.

Figure 1 displays maps of the mean SWH according to offshore and coastal definitions, and the difference between these measured along the same track. For each altimetry track, the circles of panel b and c are centred on the coordinates of the coastal location being compared. The highest mean coastal SWHs are registered along the Chilean Patagonian coast, with up to about 4 meters of average wave height (Fig. 1b), This is a notable distinction to the well-understood climatology of offshore wave heights (Fig. 1a), where the highest mean SWHs are observed in the Indian Ocean sector of the Southern Ocean[9]. These maxima correspond with the position of the southern extratropical winds, and the contribution of the eastward propagating swell on westward facing coastlines. The mid-latitudes and eastern coasts are instead characterised by smaller values.

Several wave processes exist in the nearshore zone that can contribute to differences in SWH between our defined offshore and coastal points. These processes can attenuate wave heights nearer to shore, via sheltering and depth effects. Wave heights may also increase in between offshore and coastal locations owing to local wind generated growth or shoaling effects. Wave–wave interactions and refractive processes may also modulate wave heights in this zone. Figure 1c shows that almost exclusively, coastal SWH are less than the offshore SWH, with varying degrees of coastal attenuation. To summarise and quantify the results, Table 1 shows the regional average attenuation of SWH between the defined offshore and coastal points. Most regions show a coastal attenuation of about 20%. The highest attenuation is seen in Greenland and Iceland (26%), characterised by stormy seas, but also very jagged coastline where sheltering effects will influence coastal wave climate. Only the western coast of North America and the Hawaiian archipelago show an attenuation of less than 10%.

We isolate a group of case studies in Fig. 2, where each subplot illustrates a different response between the offshore and coastal mean SWH values. Knowledge of the dominant wave direction is invaluable to best interpret the relevant acting processes. Since altimetry data do not contain information on the wave direction, we average the monthly values of wave direction available from the ERA5 reanalysis onto a 0.5° × 0.5° grid[18]. For each case study, we overlay the mean SWH obtained from altimetry (colour scale) with vectors displaying the mean wave direction.

In Fig. 2a, showing a section of the Alaska's coast, we can now resolve the sheltering effect of the island, either as a full or partial barrier, resulting in smaller SWH nearshore relative to the landward propagating swell observed on the up-wave (windward) side of the island(s). Attenuation of SWH due to the sheltering effect of islands is also seen on the global scale, where it can have influence over very large distances. For example ref. [19], reports the broad scale effects of sheltering from the Azores Archipelago on the Atlantic wave climate. In Fig. 2b, we are able to discern the depth-induced dissipation of wave energy, seen as a reduction in the altimeter measured wave height as the waves propagate across the continental shelf towards the south coast of the Australian continent. An associated refraction of wave direction is seen with small anticlockwise changes in reanalysis derived mean wave direction between the off-shelf and on-shelf locations

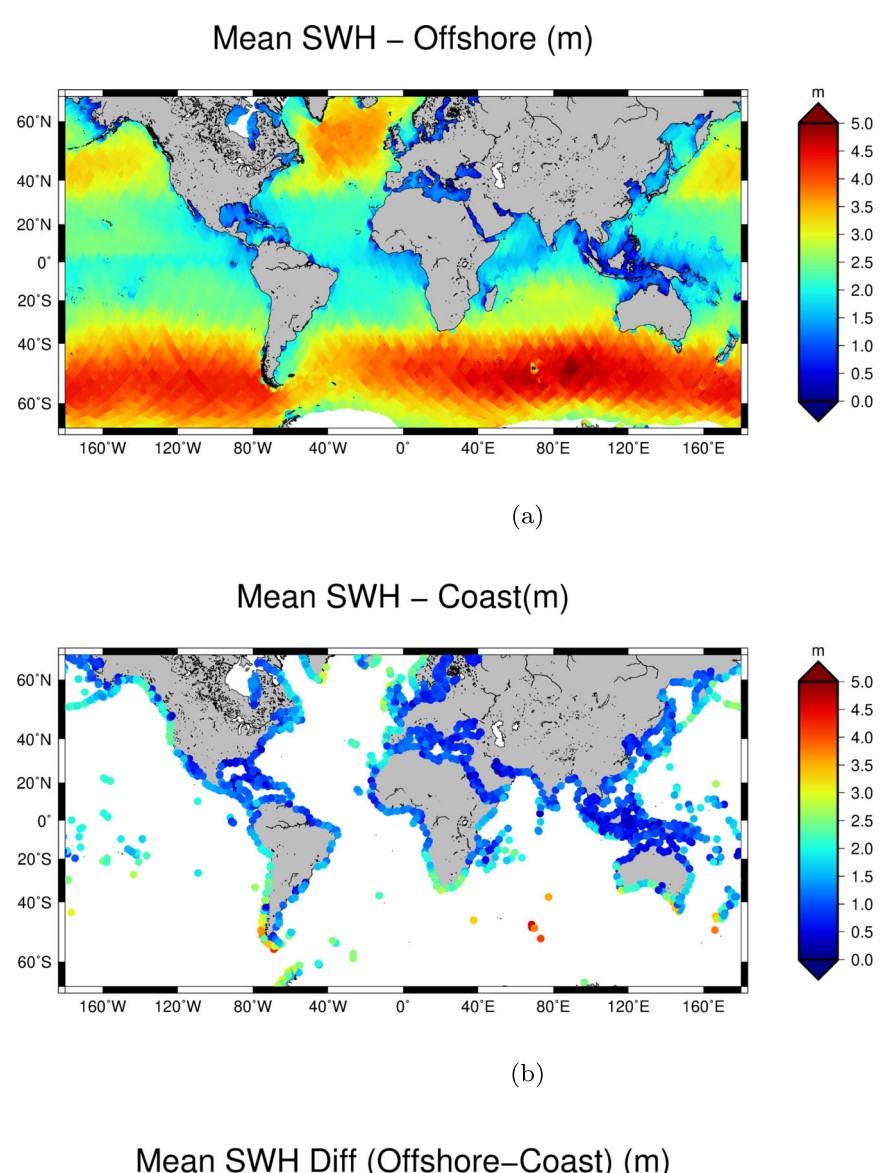

(a)

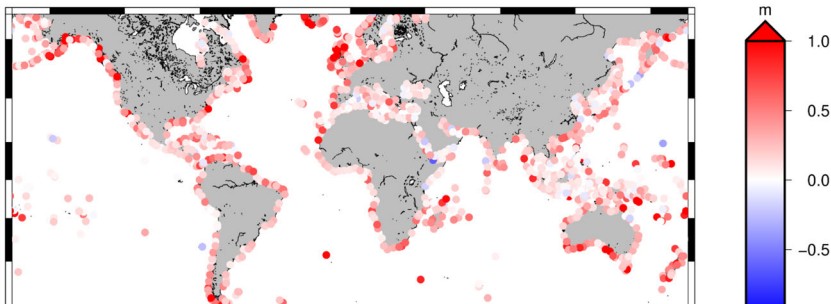

(b)

(c)

**Fig. 1 Mean Significant Wave Height (SWH).** Mean SWH from along-track satellite altimetry over the global ocean (**a**) and along the coastline (**b**). (**c**) shows the difference between the offshore and the coastal estimates.

**Table 1 Regional comparison of the mean SWH between offshore and coast.**

| Region | SWH offshore (m) | SWH coastal (m) | Diff (%) |
|---|---|---|---|
| North America (E) | 1.43 | 1.11 | 22.01 |
| North America (W) | 2.56 | 2.37 | 7.48 |
| South America (W) | 2.76 | 2.21 | 19.99 |
| South America (E) | 1.66 | 1.30 | 21.67 |
| Hawaii | 2.21 | 2.16 | 1.96 |
| Greenland and Iceland | 2.87 | 2.12 | 26.09 |
| Europe (N&W) | 2.23 | 1.78 | 22.41 |
| Africa (W) | 1.66 | 1.47 | 11.22 |
| Africa (E) | 1.63 | 1.46 | 10.53 |
| Madagascar | 1.61 | 1.28 | 20.68 |
| Asia (S) | 1.20 | 0.98 | 18.78 |
| Polynesia | 1.30 | 1.07 | 18.05 |
| Asia (E) | 1.90 | 1.56 | 18.02 |
| Australia and New Zealand | 1.91 | 1.49 | 21.87 |

The regional boundaries are reported in section Subdivision of coastal ocean.

(represented by an average 5.2° change in direction of black arrows in the corresponding locations).

As in Fig. 2a, our case study presented in Fig. 2c also displays higher mean SWH offshore. However, in contrast to Fig. 2a, we see the waves propagating offshore, suggesting the higher offshore SWH are attributable to local wind-generated growth in fetch-limited conditions.

Few locations in our dataset show no change of mean SWH between offshore and the coast, or a slight increase. One example is provided in Fig. 2d, located in Eastern Australia in the region of the Great Barrier Reef. Here, the SWH attenuation caused by the reef (visible by the bathymetric contour at −20 m depth) is counteracted by additional growth on the landward side of the reef.

Shallow–water interactions may also drive an increase in SWH, whether via shoaling or convergence of wave energy, as for example via refraction around headlands. While our observations show that on average the attenuation of SWH from offshore up to 3 km from the coast is prevalent, this does not exclude that locally the average SWH can increase in the last 3 km. This limitation might be overcome in the next years, when Delay–Doppler altimeters on repeated tracks will have acquired time series that are long enough to observe a mean behaviour. These altimeters are characterised by a better signal-to-noise ratio and along-track resolution, which could enable to fill the remaining coastal gap.

**Amplitude of the annual cycle.** Figure 3 shows the amplitude of the annual cycle of SWH in the global ocean, its coastal value found in this study and the difference between coastal and off-shore estimations (offshore-coast). Open ocean features (Fig. 3a) resemble what has been previously described by ref. [9] and ref. [10] with a gridded dataset, even if the use of along-track measurements provides less observations on a single location and no spatial interpolation with neighbouring tracks are performed in this study.

In Fig. 3b, c the amplitude of the annual cycle in the coastal regions and the difference between the offshore and the coastal estimate of the amplitude are shown.

A total of 26% of the locations show a statistically significant (black outline in figure) attenuation of the seasonality. This attenuation is largely consistent with a proportional attenuation of the mean SWHs presented in Section Mean Significant Wave Height. While there are also areas showing an amplification of seasonality in the coastal sites, the values are not statistically significant.

**Average wave energy flux.** Figure 4a provides a global view of the average wave energy flux computed with our dataset, exhibiting the expected spatial variability consistent with the mean SWH (Fig. 1a). The results agree also with the estimations of refs. [11,20] which were generated using the WaveWatch3 model[21].

Figure 4b shows the average coastal wave energy flux and Fig. 4c the difference between coastal and offshore estimations (offshore-coast). A direct comparison can be made with ref. [22], which used altimetry measurements spanning over two years to evaluate global wave resources along the coast. Their estimations are much higher and closer to the open ocean ones, due to the fact that they relied on an average of 3 along-track points (i.e., over 20 km along-track) and they could not exploit data in the last kms within the coastline. Largest differences between the offshore and coastal estimates of wave energy flux are found along the Chilean coast, on the southern tip of New Zealand, along the South East coast of Australia and in the North West coast of Europe. The distinction between the offshore and coastal energy flux representations is well observed also along the Western US Coastline. Some areas that showed offshore high energy flux in previous studies are shown here to be affected by a strong reduction of the energy flux within few kms: examples are several locations in Iceland, in the south-west coast of Australia, and along the south-east coast of South Africa.

These results are summarised in Table 2 according to the region. The most powerful waves on average are observed along the Pacific coast of the American continent (25.39 kW/m on the North American coast, and 24.20 kW/m along the South American coast). The high energy along the North American coast is notable in that this region displays the second smallest attenuation of wave energy flux from offshore to coastal (17%) after Hawaii (7%). In contrast, the relatively energetic wave climates off Greenland, Iceland, and the NW European Shelf display high attenuation from offshore to coastal values (42% and 41% respectively).

For a further check of the reliability of our estimates, it is possible to look at regions in which high resolution wave models are available. One of the regions of high interest is the Southern Australian Margin. The Australian Wave Energy Atlas[23,24] presents the wave climate around the Australian continent at a resolution of approximately 4 km and is based on a global implementation of the WaveWatch3 (v4.08) hindcast, with a series of nested high-resolution computational grids in the Australian and South Pacific region. This dataset enables comparison of the cross-shelf gradients of wave energy flux in this region. The agreement between these model results and our derivation from the coastal altimetry data, shown in Fig. 5a, is quantified in Fig. 5b, where the mean and standard deviation of the differences between altimetry-derived and model-derived results is plotted with respect to the distance to coast, binned every 3 km. The mean bias is below 1 kW/m in the first 60 km from the coast, with a maximum standard deviation of about 5 kW/m close to the coast. Further away, altimetry tends to slightly overestimate the flux, but the mean bias is on average below 4 kW/m regardless of the distance to coast, i.e., less than 13% of the modelled wave energy flux.

**Discussion**

The use of reprocessed time series of coastal altimetry data provide the chance to observe the interaction between waves,

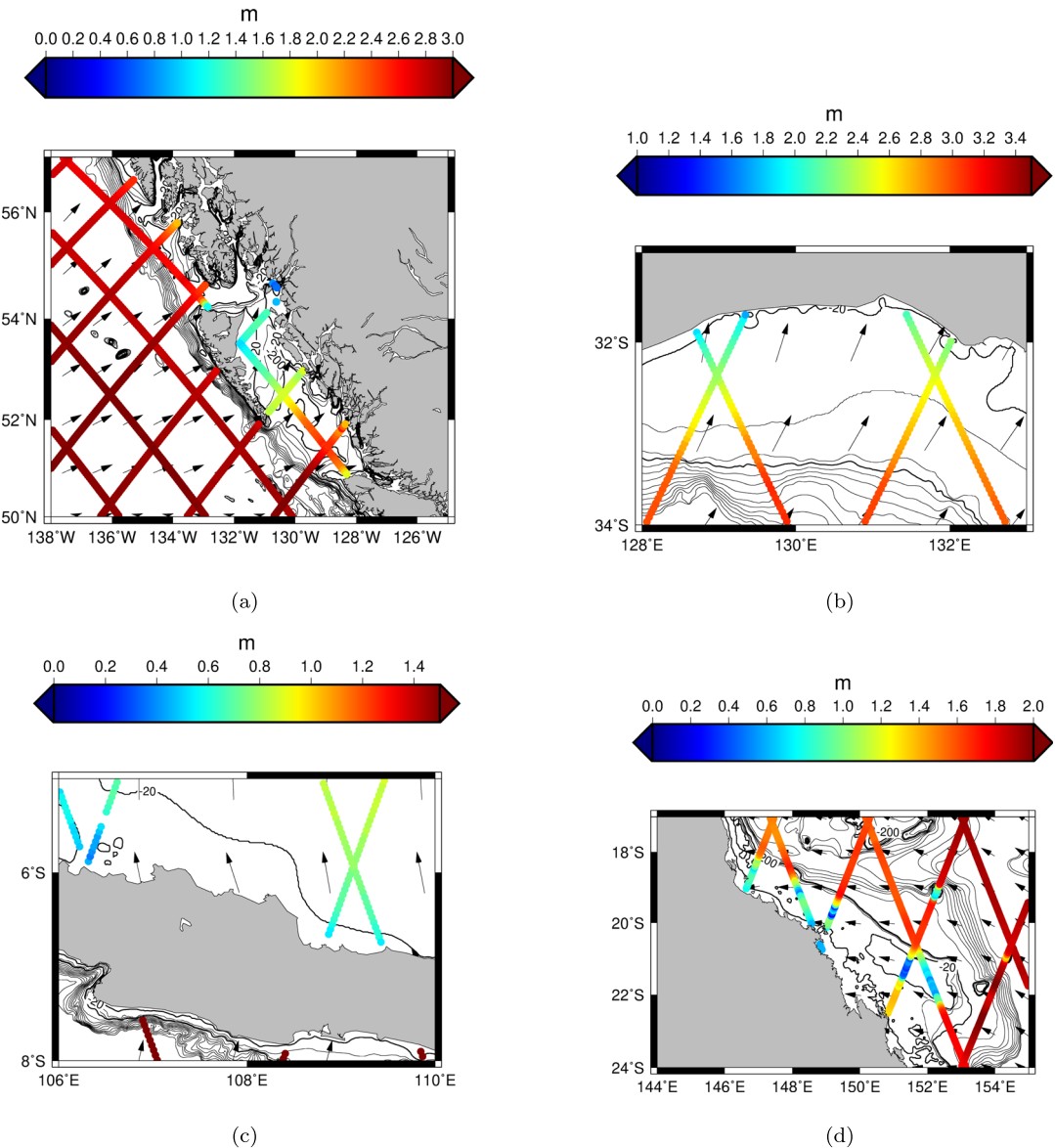

**Fig. 2 Coastal changes of mean Significant Wave Height (SWH).** Examples of coastal changes of mean SWH along the altimetry tracks (colour scale) in Alaska (**a**), South Australia (**b**), the island of Java in Indonesia (**c**) and in Great Barrier Reef region of East Australia (**d**). Bathymetry contours are plotted at intervals of 60 m from −20 m until −200 m depth and every 200 m until −2000 m depth. The mean wave direction computed from the ECMWF ERA5 reanalysis is shown with black arrows.

bathymetry, and coastlines in terms of SWH. Besides the common understanding that SWH is decreased in the coastal zone, the study quantifies the attenuation of mean state, seasonality, and wave energy flux at an unprecedented resolution that could so far only been achieved using dedicated high-resolution models for regional and local downscalings[25].

The results are summarised in their global statistics in Fig. 6, where the parameters are shown as a ratio against the value at 30 km from the coast (defined as offshore in this study). The polynomial fit indicates that the mean SWH at 3 km from the coast is on average 22% smaller than offshore, the amplitude of the annual cycle is reduced in the same distance on average by 14% and the average energy flux loses 38% of its offshore value. The global coastal attenuation is verified with a confidence level of 95% for both mean SWH and average energy flux. This is not true for the amplitude of the annual cycle, whose difference between coastal and offshore values has a wider spread.

While dedicated regional high-resolution models are able to take into account the attenuation seen with satellite altimetry, studies of global wave power up to now, including the assessment of the World Energy Council[26] have typically been defined using models or reanalysis with validation relative to offshore satellite altimetry data. The resolution of these models may be high for regional applications (e.g., of order hundreds of metres[27]). At global scale, the resolution of these models is typically in the range of 0.25°[25] to 0.5°[11]. The wave energy generation systems are typically planned to be placed near the shore or in depth ranges of 30–50 m in the offshore cases[24]. Given the global observational representation of the coastal attenuation provided in this study, studies of global wave power shall be therefore updated using the latest models at higher resolution.

Finally, this study shows the level of accuracy that reprocessed satellite altimetry offers to describe the coastal wave climate in terms of SWH. Our dataset is unprecedented in presenting altimeter wave height data near to the coast at a

### SWH Annual Cycle Amplitude – Offshore (m)

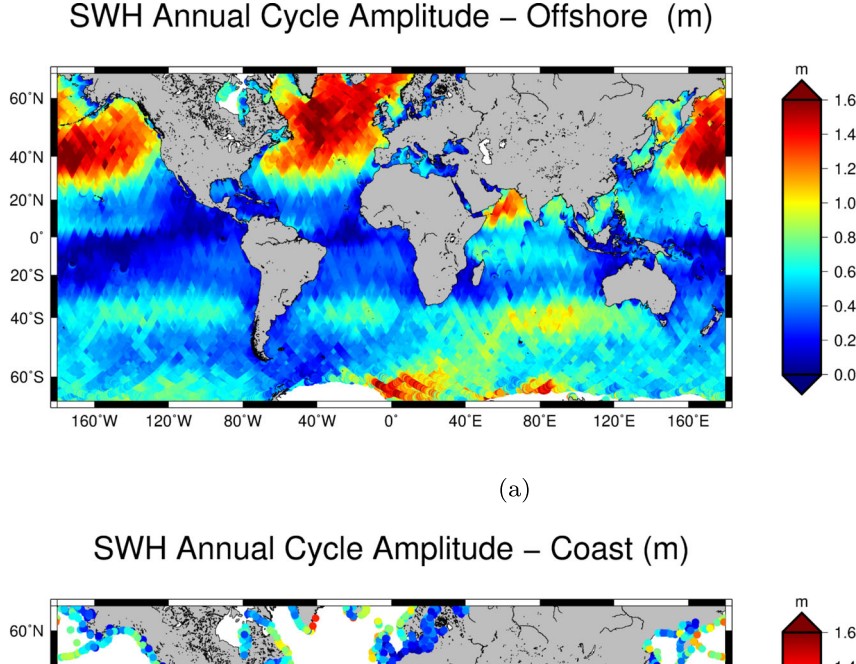

(a)

### SWH Annual Cycle Amplitude – Coast (m)

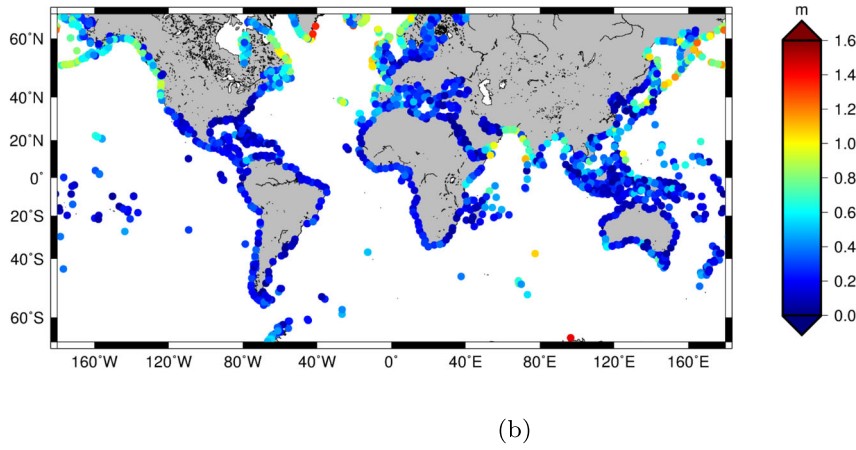

(b)

### SWH Annual Cycle Amplitude Diff (Offshore–Coast) (m)

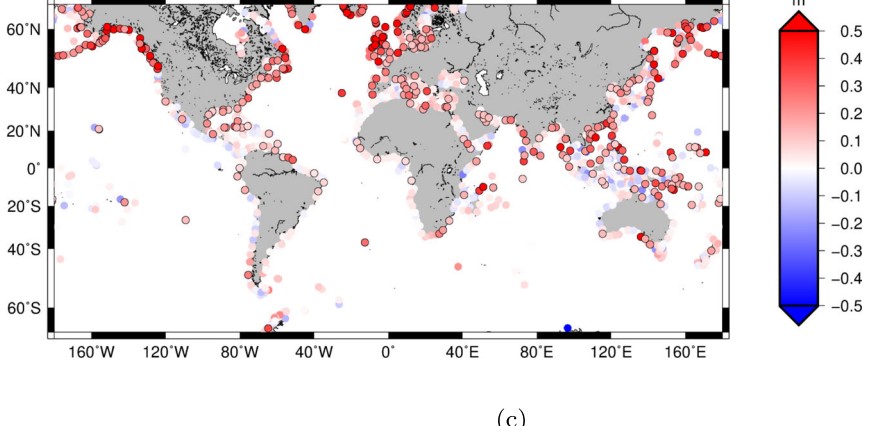

(c)

**Fig. 3 Amplitude of the annual cycle.** Amplitude of the annual cycle computed for significant wave height (SWH) time series from along-track satellite altimetry over the global ocean (**a**) and along the coastline (**b**). **c** shows the difference between the offshore and the coastal estimate of the amplitude. Statistically significant differences are marked with a black contour. The point is marked as significant if the absolute value of the difference between the offshore and the coastal amplitude is higher than its uncertainty. Uncertainties are computed as described in section Computation of mean SWH and annual cycle.

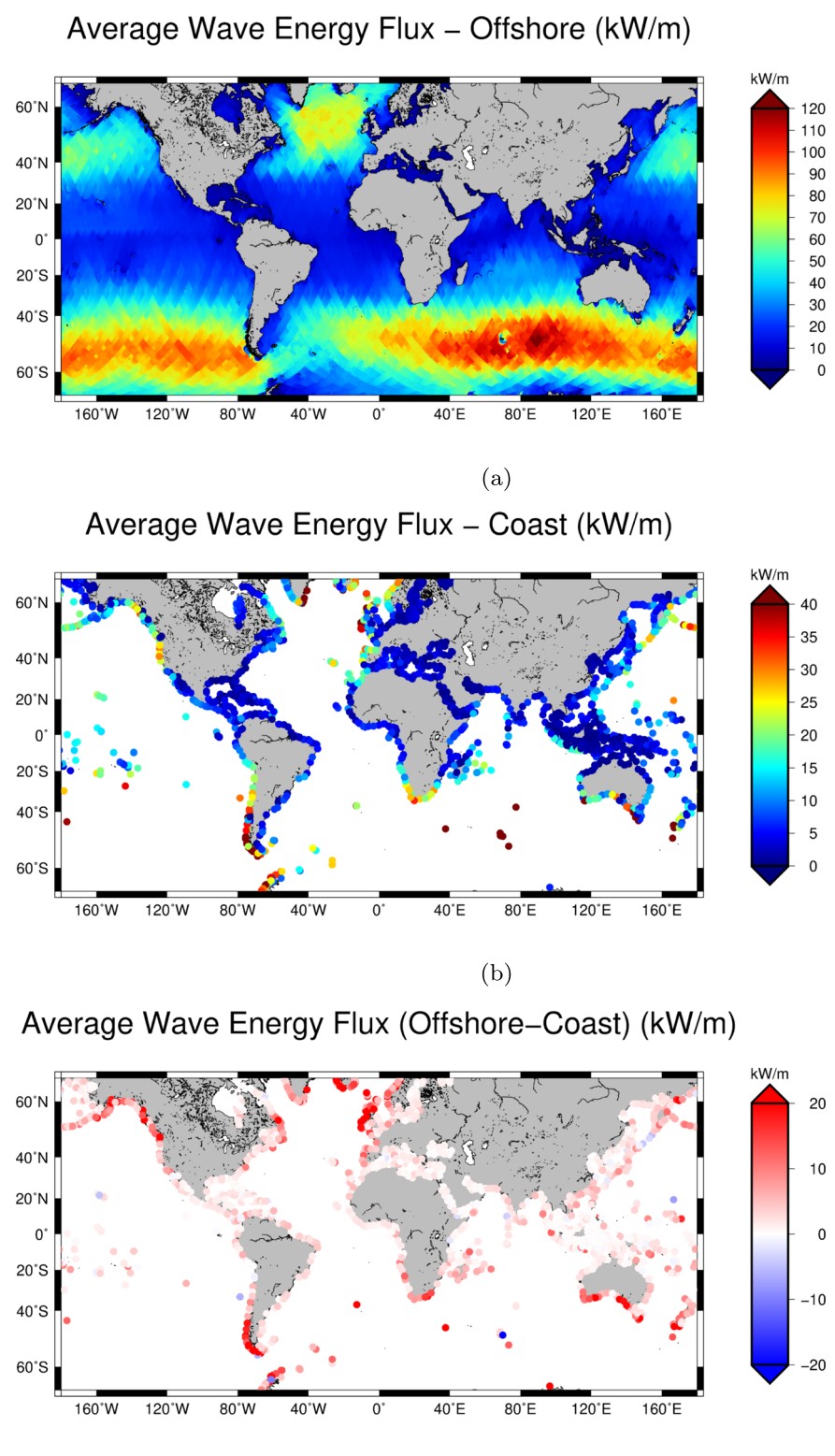

**Fig. 4 Average wave energy flux.** Average wave energy flux from along-track satellite altimetry over the global ocean (**a**) and along the coastline (**b**). (**c**) shows the difference between the offshore and the coastal estimates.

global scale. This opens possibilities for future global analyses seeking to quantify near coastal wave transformations across a full distribution of shelf environments. The short time series with respect to the variability of this quantity still hampers the estimation of significant trends[12]. Future efforts in this sense, which are planned for example in the framework of the European Space Agency's Sea State Climate Change Initiative[28], shall focus on a dedicated irregular coastal gridding in order to increase the sampling while avoiding rough interpolation with offshore data.

**Table 2 Regional comparison of the average wave energy flux (WEF) between offshore and coast.**

| Region | WEF offshore (kW/m) | WEF coastal (kW/m) | Diff (kW/m) |
|---|---|---|---|
| North America (E) | 8.11 | 4.81 | 3.30 |
| North America (W) | 30.62 | 25.39 | 5.24 |
| South America (W) | 37.10 | 24.20 | 12.91 |
| South America (E) | 9.34 | 5.93 | 3.40 |
| Hawaii | 19.75 | 18.61 | 1.14 |
| Greenland and Iceland | 42.15 | 24.58 | 17.57 |
| Europe (N&W) | 28.44 | 16.87 | 11.57 |
| Africa (W) | 12.29 | 10.10 | 2.20 |
| Africa (E) | 11.60 | 8.66 | 2.95 |
| Madagascar | 11.00 | 7.48 | 3.52 |
| Asia (S) | 6.10 | 4.09 | 2.00 |
| Polynesia | 7.16 | 5.27 | 1.88 |
| Asia (E) | 16.46 | 11.76 | 4.70 |
| Australia and New Zealand | 19.50 | 12.61 | 6.89 |

The regional boundaries are reported in Section Subdivision of coastal ocean.

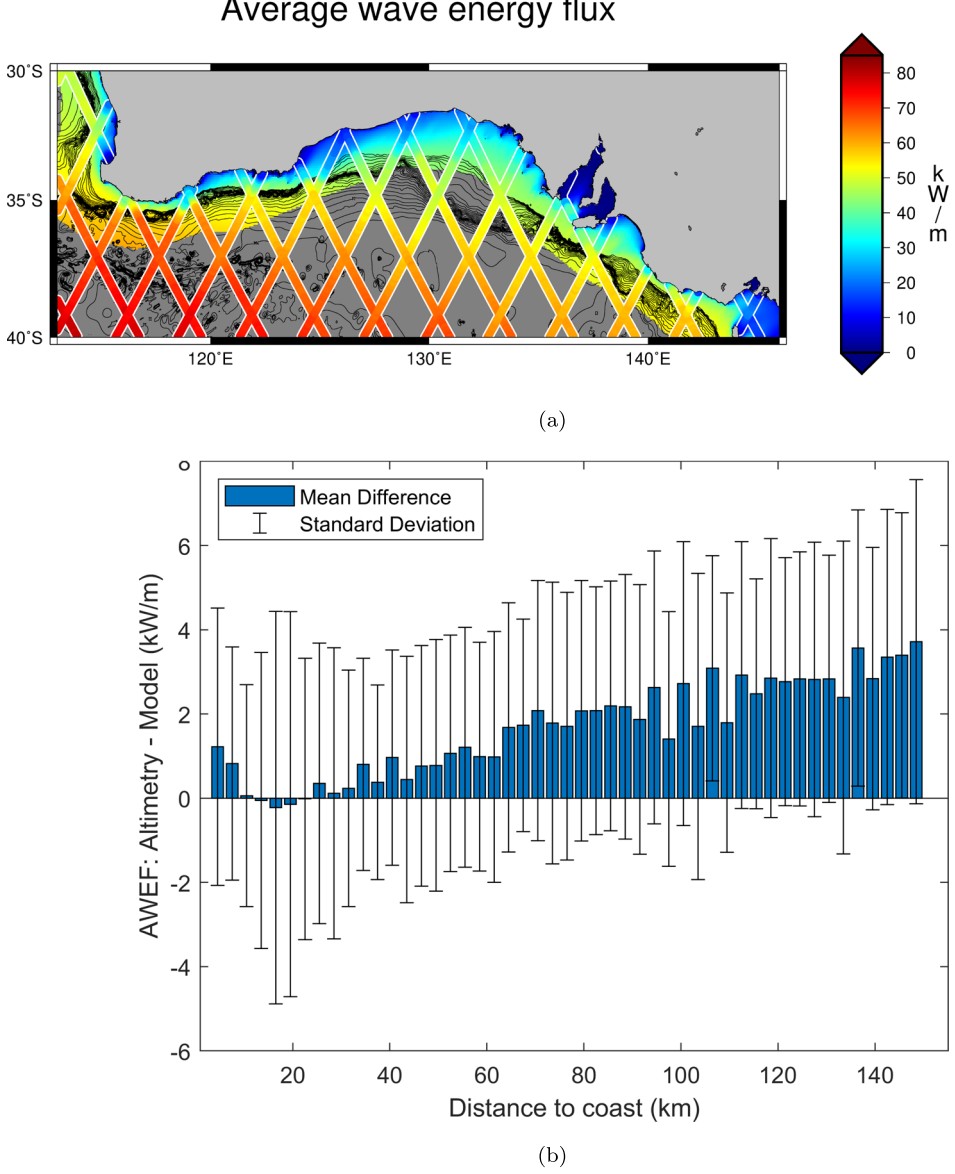

**Fig. 5 Coastal average wave energy flux in South Australia.** Average wave energy flux (AWEX) along the altimetry tracks in South Australia superimposed over the 50th percentile of wave energy flux computed using model data[23] (**a**). Mean difference (blue bars) and standard deviation of the differences (error bars) between colocated altimetry and model locations with respect to the distance to coast binned every 3 km (**b**).

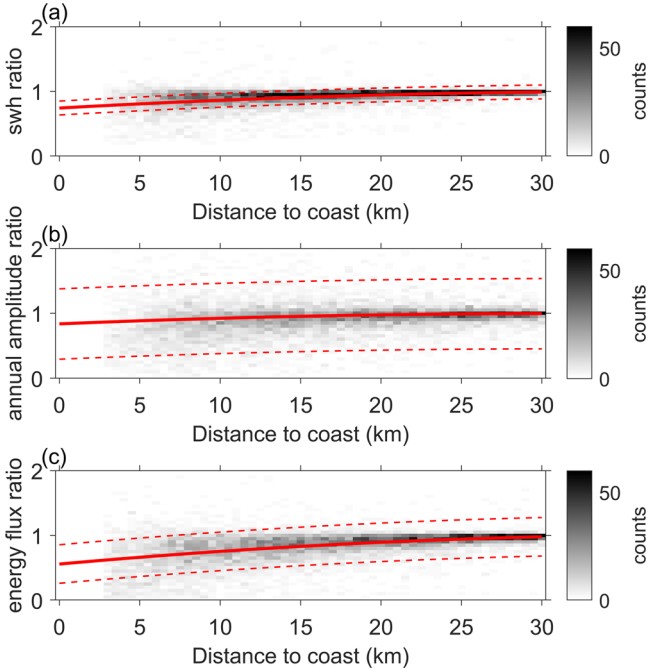

**Fig. 6 Global coastal attenuation.** Density plots of the ratios between wave parameters computed in the coastal zone over the globe and the corresponding parameter computed offshore. The parameters considered are the mean Significant Wave Height (SWH) (**a**), the amplitude of the annual cycle (**b**), and the average wave energy flux (**c**). A second-degree polynomial is fitted to the data and plotted in red. The 95% confidence interval of the fit is shown with red dashed lines.

## Methods

### Processing of satellite altimetry data

*High-frequency retracking.* We analyse satellite altimetry data coming from the Sensor Geophysical Data Records (SGDR) of Jason-1 and Jason-2 missions, from July 2001 to January 2016. The SGDR contain returned radar echoes, called waveforms, at a 20-Hz rate (corresponding to a distance of about 350 m). Routinely in the standard product, a functional form (the Brown-Hayne model) is fitted to the waveform in a process called retracking. The SWH is directly estimated from the Brown–Hayne model[29,30].

Several waveforms in the last 0 to about 20 km from the coast differ from the theoretical Brown–Hayne shape due to the inhomogeneity of the backscatter of the illuminated area in the coastal zone. For this reason, data in the coastal strip are routinely flagged or discarded. In this study we use instead SWH data that are retracked with the ALES algorithm, which only considers a portion of the waveform in order to recover data in the coastal zone, while maintaining the quality of the retrieval in the open ocean as well.

In the ALES retracker, the retracking of each waveform is performed in two steps. A first step looks at the rising portion of the waveform (called leading edge) and provides a rough estimate of SWH from the slope of that portion. This estimate is then entered into an algorithm that selects the sub-waveform (i.e., sets the width of the fitting window over which a fitting is performed in the second step). The dependence on the SWH is necessary to maintain the same level of precision achievable in the open ocean using a full-waveform retracker, given the direct relationship between sea state and noise of the retrieval.

A full description of the ALES retracking procedure is provided in ref. [15]. The SWH detection with the ALES retracker was validated against buoys in[16], which confirmed that ALES is able to extract meaningful retracked parameters up to about 3 km from the coast, which is also the limit of validity adopted in this study.

*Low-frequency averages.* In order to decrease the noise of the high-frequency retrievals, standard altimetry data are routinely averaged at a 1-Hz rate (approximately one measurement every 7 km).

We briefly recall the procedure used to average ALES retracked SWH to generate 1-Hz estimations[16]. A check is performed in order to eliminate outliers on every block of 20 high-rate values X: the median value and the scaled median absolute deviation ($\widehat{\text{MAD}}$) are computed. Each estimation x is considered valid if:

$$x < \text{median}(X) + 3 \times \widehat{\text{MAD}}(X) \tag{1a}$$

or

$$x > \text{median}(X) - 3 \times \widehat{\text{MAD}}(X) \tag{1b}$$

where

$$\widehat{\text{MAD}}(X) = 1.4286 \times \text{median}(|X - \text{median}(X)|) \tag{1c}$$

The scaled $\widehat{\text{MAD}}$ uses the factor 1.4286 and is approximately equal to the standard deviation for a normal distribution. Statistics based on the median are more robust and suitable for outliers detection and have been already applied to satellite data[31]. Once the outliers have been excluded, the median of the remaining points is computed in order to generate the 1-Hz estimation.

It has to be noted that the 1-Hz SWH value along the track is nominally located at the centre of a segment of 20 20-Hz measurements and therefore is affected by the SWH retrievals located up to about 3.5 km before and after the nominal along-track point. In this study, a 1-Hz average is computed only if after the outlier procedure there are at least six valid 20-Hz measurements in the 1-Hz block.

The SWH estimations for Jason-1 and Jason-2 are corrected using the instrumental corrections, as described in refs. [32] and[33].

*Cross-calibration of the missions.* Although Jason-1 and Jason-2 were very similar missions aimed at the continuity of the records, biases in the retracked parameters between different missions are common and must be taken into account in a cross-calibration exercise[34]. For this purpose, following previous studies focused on the standard products[35], we exploit the Jason-1/2 tandem mission, with the altimeters flying the same track 54s apart (cycles 1–20 of Jason-2 and 240–259 of Jason-1). The bias is computed on each 20-Hz location.

We show the results in Supplementary Fig. 1 for different sea states. Biases between the SWH from the two altimeters are likely to be caused by the treatment of the Point Target Response in the Brown–Hayne model, which approximates it with a Gaussian function[33]. Nevertheless, the bias is two orders of magnitude smaller than the SWH parameters analysed in this study (annual cycle, mean SWH), which are on the order of meters. Given the relatively small differences found in dependence with the sea state and since spurious small drifts in trends do not affect the results in this study, we limit our cross-calibration to the application of a constant bias obtained as median of the available comparisons, i.e., we subtract 0.03 cm to every SWH measurements from Jason-2.

**Computation of mean SWH and annual cycle.** In order to compute along-track 1-Hz averages to create a time series, data points along the satellite tracks have to be collinear: it is necessary to have measurements at the same geographical location for each cycle. Nominal tracks are therefore created for this study using the reference orbits, neglecting the across-track displacement of different passes along the same track, which is normally less than 1 km. Each interval between con-secutive 1-Hz data points is divided in order to obtain 20 equidistant nominal locations, along which the SWH data for each cycle is then linearly interpolated. As a result of this process, at each lat-lon couple corresponds a time series with a record per each cycle. The mean SWH field is then simply the mean SWH of each of this time series.

To estimate the annual cycle, once the cycle-by-cycle time series are adapted into monthly averages, a harmonic analysis of the time series is performed. The analysis consists of modelling the sea level variability as the sum of a constant, a linear term, and a sinusoid wave with an annual frequency. The unknowns (parameters) of this model are the constant term, the slope of the linear term and the amplitude and phase of the sinusoid. Amplitude and phase of the annual frequency are not independent parameters, since they are estimated through the same fit, according to the following model:

$$y = A + Bx + C\cos 2\pi xf + D\sin 2\pi xf \tag{2}$$

where A-D are the coefficients to be estimated and f is the annual frequency. The amplitude (Am) and phase (Ph) of the annual signal are then computed as follows:

$$\text{Am} = \sqrt{C^2 + D^2} \tag{3a}$$

$$\text{Ph} = \begin{cases} \arctan\left(\frac{D}{C}\right) & \text{if } C < 0 \\ \arctan\left(\frac{D}{C}\right) + \pi & \text{if } C > 0 \end{cases} \tag{3b}$$

Since we are dealing with geophysical time series, in order to correctly express the uncertainty on the estimated annual cycle it is necessary to account for autocorrelation and therefore Feasible Generalised Least Square (FGLS) methods are used instead of the standard Ordinary Least Squares. We use the Prais-Winsten (PW) estimator[36], which applies a transformation to the dependent and independent variables in order to transform the problem into one that respects the Gauss-Markov hypothesis. The PW estimator is applied in the present study iteratively. Given a set of independent variables X, observations Y, error $\epsilon$ and parameters to be estimated $\beta$, the method finds the term $\rho$ that expresses the correlation of the residuals. The steps followed are:

1. Ordinary least squares estimation of the model $Y = \beta X + \epsilon$

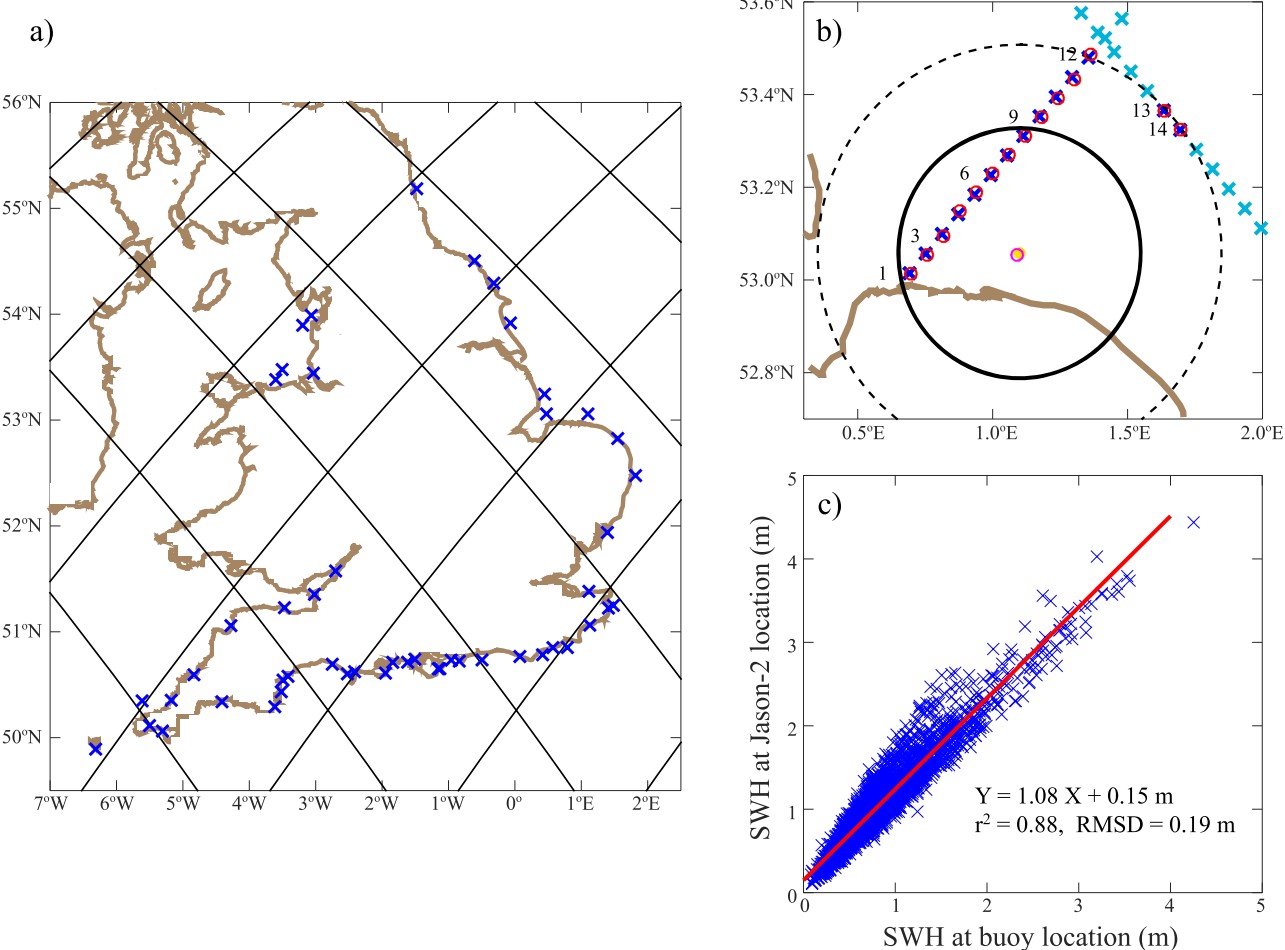

**Fig. 7 Regional validation against coastal buoys. a** Jason-2 tracks overlaid on map of the 50 coastal wave platforms curated by the Channel Coastal Observatory. **b** Zoom over region of Blakeney Overfalls buoy (orange star) off East Anglia, with nominal points along two Jason-2 tracks shown by light blue crosses. The concentric circles indicate 50 km and 30 km from the buoy, with dark blue crosses indicating those within 50 km. The red and magenta circles indicate the centres of the corresponding model points. **c** Linear regression of Significant Wave Height (SWH) values from reanalysis at altimeter point 9 (y-axis) against reanalysis values at the buoy location.

2. Ordinary least squares estimation of the model $\epsilon_t = \rho\epsilon_{t-1} + e$ in order to estimate the parameter $\rho$, which is related to the first order autocorrelation of the residuals

3. Ordinary least squares estimation of the transformed model $Q = \beta Z + e$, in which

$$
Q_t = \begin{cases} Y_t\sqrt{1-\rho^2} & t \equiv 1 \\ Y_t - \rho Y_{t-1} & t \equiv 2 : \text{end} \end{cases} \tag{4}
$$

and

$$
Z_t = \begin{cases} X_t\sqrt{1-\rho^2} & t \equiv 1 \\ X_t - \rho X_{t-1} & t \equiv 2 : \text{end} \end{cases} \tag{5}
$$

where $t$ is the time index of the observations. The procedure is iterated until $\rho$ converges to a value close to zero and therefore the errors of the transformed model are no longer autocorrelated.

**Validation of the mean parameters against buoys.** We use the global network of buoys provided by the Copernicus Marine Environment Monitoring Service (CMEMS, marine.copernicus.eu) to validate satellite altimetry data. Along a track, altimetry data are available every 10 days, as opposite to the sub-daily measurements from buoys. A validation is therefore necessary to test quality of the estimation of mean parameters despite this sampling. The validation is performed by comparing the time series of SWH along consecutive satellite passes with the time series generated by buoy measurements. Overlapping time periods between buoys and altimetry are used. Buoys containing less than 2 years of data are discarded.

We compare the performances of the ALES SWH against the standard geophysical data records (GDR) in terms of Pearson correlation coefficient with

respect to the buoys. We select all altimetry points between 30 and 3 km from the coast and within 30 km from a buoy. Given corr(buoy,ALES) the correlation between SWH time series from ALES and buoys, and corr(buoy,standard) the correlation between SWH time series from GDR and buoys, Supplementary Fig. 2 shows the difference corr(buoy,ales)—corr(buoy,standard) with respect to the distance to coast of the along-track location. In 85% of the cases, the dataset used in this study has a higher correlation than the one achieved by a standard product. This confirms previous regional validation efforts of the ALES SWH estimations[16].

Secondly, we check the suitability of using along-track data to derive mean SWH and annual cycle. The standard procedure followed in the previous literature consists in gridding the data to average more tracks together within boxes of at least 1° spacing in latitude and longitude. This strategy is not suitable for this study, since doing so would smooth the differences at the coastal scales that we want to study. Therefore, we compare mean SWH and amplitude of the annual cycle computed with the data from the buoys with the same variables computed using the closest point of the satellite track, provided it is not located further than 30 km from the buoy. Given that in the coastal proximity the SWH changes much more rapidly, as shown in this study, we restrict this distance to 15 km for buoys located closer than 30 km to the coastline. Using these criteria, 51 altimetry-buoy couples were found, out of which 11 featuring coastal buoys.

The results show values of correlation and slope close to 1 in both mean and annual cycle amplitude of SWH (see Supplementary Fig. 3). Notably, the performances of the coastal data (i.e., using buoys located closer than 30 km of the coast, highlighted in red) do not differ from the data offshore. Despite the short overlap of some altimetry-buoy couples, only 12% show a significant difference in the amplitude of the annual cycle.

*Regional validation against coastal buoys.* A downside of our global validation approach is that the comparison between the buoy and the satellite track points may not always be valid, because the buoy and altimeter measurements could be in

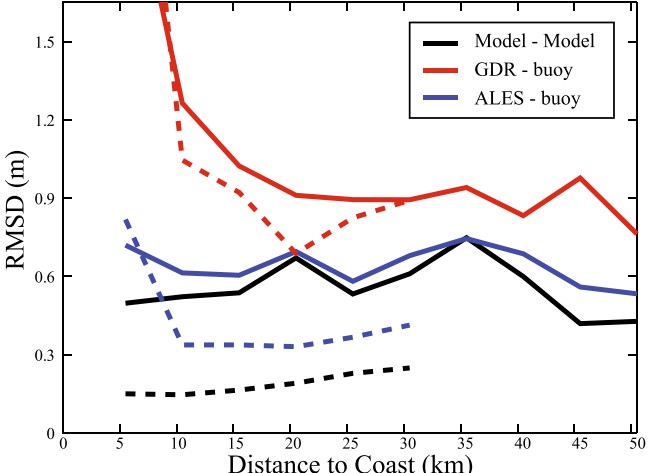

**Fig. 8 Coastal comparison between buoys, altimetry, and reanalysis.** Root Mean Square Difference (RMSD) of observations at altimeter point and at buoy location, with respect to the distance to coast binned every 5 km. Black lines indicate comparison of reanalysis at the two locations; red and blue are for standard Geophysical Data Records (GDR) and this study (ALES) respectively compared with the buoys. Solid line shows all comparisons; dashed line shows results for pairings deemed highly coherent i.e., for which the reanalysis comparison has $r \geq 0.95$ and RMSD $\leq 0.3$ m.

very different coastal environments (see Fig. 3 of ref. [37]), leading to significant SWH spatial gradients as the coast is approached. For this reason, we provide a further regional validation adopting the approach of ref. [38] in using hindcast outputs from a high-resolution wave model to assess the contribution due to spatial variation in the wave field.

As hindcast, we use the NWSHELF_REANALYSIS_WAV_004_015 product from CMEMS. As indicated in the product, this is a reanalysis based on the WaveWatch3 model and the North-West Shelf configuration is based on a two-tier Spherical Multiple Cell grid mesh (3 and 1.5 km cells). The product is further referred to as WW3 in this section. As a buoy database we use the Channel Coastal Observatory containing data from 50 platforms around the English coast, of which we identified 13 as being within 50 km of a Jason track and having several years of overlap with the Jason-2 record on those tracks. We then identify all the nominal 1-Hz altimeter points within 50 km of the in situ measurements, and compared both the GDR and the ALES-retracked values against the reference.

The fine-resolution grid of WW3 enables us to use reanalysis data within ~11 km of the locations of buoy and altimeter measurements (Fig. 7). A linear regression analysis is performed between WW3 fields at the two locations (Fig. 8) to give the best-fit line with correlation $r$ and the root mean square difference (RMSD).

Similar regression analysis is then performed for the buoy and altimeter observations, first considering all the matchups within 50 km (Fig. 8). The RMSD for these accepted pairings is partitioned into distance from coast using 5 km wide bins. The values for ALES (~0.6 m in blue) are only slightly greater than the spatial variation seen within WW3, whereas those for the GDR (~0.8 m, in red) are significantly larger and increase markedly within 15 km of the coast. Restricting the selection to buoy-altimeter pairings for which WW3 values at the two locations are highly coherent ($r \geq 0.95$ and RMSD $\leq 0.3$ m) decreases the number of matchups from 160 to 44, and leads to a further reduction in the RMSD values for ALES to ~0.4 m until within 8 km of the coast, but has little impact on the perceived accuracy of the GDR values. We also note that the median value of the RMSD for ALES is 0.35 m even for the bin 3–8 km and the mean in the plot has a larger RMSD due to three matches with a higher discrepancy. Therefore we conclude that the ALES-derived estimates are significantly more accurate than those on the GDR, with the RMSD for the former being dominated by the spatial changes in wave height rather than error in the retracker algorithm.

**Computation of the average wave energy flux.** Wave energy flux has a dependency on wave period and wave heights. Altimeters have limited ability to provide estimates of wave period[39], and consequently we follow the approach of others (e.g.,[40]) by supplementing the altimeter derived wave heights with reanalysis derived wave periods to determine the wave energy flux. Here, zero-crossing wave periods are obtained from the ERA5 reanalysis at a time resolution of 3 hours and spanning the same time period as the altimetry observations.

We first compute the instantaneous wave energy flux for each satellite cycle at each location, using the following relationship[41]:

$$P = \frac{\rho g^2}{64\pi} \text{SWH}^2 T_e \quad (W/m) \tag{6}$$

where $\rho$ is the sea water density (1025 $kg/m^3$). The energy period $T_e$ can be derived from the zero-crossing period using the relationship:

$$T_e = \alpha T_z \tag{7}$$

as a value for $\alpha$ we use 1.18, in accordance to ref. [42]. We thus accept the assumption of a constant spectral shape, which introduces some uncertainty. Its effect is nevertheless negligible, because the error, when estimating $\alpha$, is an order of magnitude smaller than that for the effects of $T_z$ and SWH[13].

By using Eq. (6), we accept the following assumptions:

- The equation is valid with a deep-water assumption, which might not be true for our definition of coastal zone, particularly for swells
- Given that statistics are based on reanalysis data on a $0.5° \times 0.5°$ grid, we are assuming that the wave period remains constant when approaching the coast. While the waves conserve their period when approaching shallow water, waves can be also generated locally.

Even if small changes in the period in the shallow waters can occur, their influence is much smaller than the changes in SWH also in quantitative terms in the equation[43]. After an estimation of the mean wave energy flux is computed at each collinear 1-Hz location for each satellite cycle, the mean of each time series is then computed, as for the SWH.

A verification of this computation in absolute terms is provided regionally against model data in Fig. 5.

**Errors due to deep water assumption.** The adoption of equations adapted for intermediate and shallow water requires knowledge of the wavelengths and the bathymetry. In order to understand the error that we are committing in using the deep water assumption, we use the bathymetry from GEBCO2020 and the wave period from ERA5 to derive an estimation of the wavelength $L$ following the parameterisation of ref. [44].

Following the linear wave theory and considering a complete spectrum, the wave energy flux is:

$$P = E c_g \tag{8}$$

where $c_g$ is the group velocity and:

$$E = \frac{1}{16} \rho g \text{SWH}^2 \tag{9}$$

Using the parameterised mean wavelength $L$ and considering the depth $d$, we distinguish between deep ($\frac{d}{L} > = 0.5$), shallow ($\frac{d}{L} < = 0.04$), and transitional ($0.04 < \frac{d}{L} < 0.5$) waters and compute the corresponding group velocity $c_g$ following ref. [45].

We notice that a downside of this approach is that, by using Eq. (8), we are associating a group velocity to a single characteristic period of an irregular wave field, although the group velocity is a function of a specific frequency of regular waves. This is a commonly applied simplification used where available wave information is limited to bulk wave parameters (such as SWH and mean wave periods)[46].

Supplementary Fig. 4 shows the 98th percentile of the differences between the computation of the average energy flux using the deep water assumption and the approximate solution considering shallow and intermediate waters. The data are binned every km according to their distance to the coast. We conclude that by extending the deep water assumption in all our domain of study we are committing an error that does not exceed 1.1 kW/m.

**Subdivision of coastal ocean.** In order to provide regional estimates of mean SWH and average wave energy flux, we aggregate the coastal and offshore locations of the altimetry tracks according to several sub-regions, following the grouping proposed by[11]. The regions and their naming are reported in Supplementary Fig. 5.

## Data availability

The processed data that support the findings of this study are available in SEANOE with the identifier https://doi.org/10.17882/80341. All dataset used in this study to produce the results are freely available as indicated in this section. Sensor Geophysical Data Records (SGDR) for Jason-1 and Jason-2 missions were downloaded from the following sources: ftp://avisoftp.cnes.fr/AVISO/pub/jason-1/sgdr_e/, ftp://avisoftp.cnes.fr/AVISO/pub/jason-2/sgdr_d/ 1-Hz significant wave height data reprocessed with ALES used in this study are stored in https://openadb.dgfi.tum.de/en/data_access/. Data from the ECMWF ERA5 reanalysis were obtained from https://cds.climate.copernicus.eu/cdsapp#!/dataset/reanalysis-era5-single-levels?tab=overview. Data from the Australian wave energy atlas model along the Southern Australian coast are extracted from the following: https://www.nationalmap.gov.au/renewables/#share=s-sTv1VCxfENCdHe2g. Reanalysis data for the North West European Shelf are available from the Copernicus Marine Environment Monitoring Service (CMEMS). The product name is

NWSHELF_REANALYSIS_WAV_004_015. Data from the Channel Coastal Observatory were provided by the National Network of Regional Coastal Monitoring Programmes of England via www.coastalmonitoring.org. The code developed to provide the results presented in this study is stored and available on request.

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

## Acknowledgements

We thank Jean-Francois Piolle (Ifremer), Guillaume Dodet (Ifremer), Andreas Sterl (Royal Netherlands Meteorological Institute), Jean Bidlot (ECMWF), and David Griffin (CSIRO) for the useful discussion and the guidance about the external data sources. Marcello Passaro also thanks Paolo Cipollini (ESA), whose supervision was of great importance in the development of the ALES retracking algorithm. The GMT software

system was used for producing the maps[47]. Francesco Nencioli (CLS) developed the concept of model-derived coherent areas used in the validation.

## Author contributions

M.P. conceptualised and designed the study and wrote the manuscript. M.H. contributed to writing the manuscript concerning the interpretation of the results and is supported by the Australian Commonwealth National Environmental Science Program Earth Systems and Climate Change Hub and the Integrated Marine Observing System. G.D.Q. performed the regional validation against coastal buoys. M.P. is the author of the full software code used in this study, except for the code used for Section Regional validation against coastal buoys, and of the ALES retracking algorithm. D.D. initiated and coordinated the collection and organisation of the altimetry dataset. C.S. and D.D. are responsible for the altimetry database organisation and the data structure used by ALES. F.S. provided the basic resources making the study possible and coordinates the activities of the research group at DGFI-TUM. All authors read and commented on the final manuscript.

## Funding

## Competing interests

The authors declare no competing interests.
