## [Peer Review File · Nature Communications]

Editorial Note: Parts of this Peer Review File have been redacted as indicated where no third party permissions could be obtained.

Reviewers' comments:

Reviewer #1 (Remarks to the Author):

General comments:

The ALES algorithm is applied to characterise the ocean waves attenuation in the coastal zone, from altimetry data (Jason-1 and Jason-2 missions). The paper gives a useful observation of the significant wave height and energy flux at a global scale both offshore and nearshore. Such a coverage can only be achieved by satellite altimetry. The application of the ALES retracker algorithm, detailed in previous papers from the same lead author, gives more reliable data close to the coast than previous altimetry data. Therefore this presents a potentially wide interest for a range of applications, such as coastal erosion/protection, model calibration, wave energy site development, or coastal maritime traffic, which justifies its publication.

The paper will influence the wave resource analysis because it solves one of the major drawback of altimetry data (unreliability of altimetry data near the coast), and it applies the method to a worldwide screening of coastal wave energy.

Overall it is an good application of the ALES retracker algorithm to estimate the global coastal wave height and energy flux.

Detailed comments:

- The paper is articulated around two different parts presenting the methods and the results. The paper reads well, although language and clarity could be improved in some cases. The section on methods better written and structured, but I was surprised to see that section after the results section. Not sure why the manuscript was structured that way, as it seems to me more logical to present the methods before the results and conclusions.

Is it possible to restructure the paper by putting section 4 before sections 2 and 3?

- Line 21 – “more in general” should better read “more generally”

- Line 91-93 – I found the illustrative example of the Azores and Kerguelan archipelagos a bit weak, as it is not easy to spot it in Figure 1a, due to the scale and resolution of the figure. Could it be a more obvious example (e.g. Cape Horn?) ?

- Figures 2:

- 1) axis and colorbar labels are too small. Could you increase their font size or enlarge the figures?

- 2) although it can be deduced from the longitude and latitude coordinates (and for most graphs the coastlines configuration), it would be good to name the locations of each plot, either in a plot title, or in the caption. For instance, readers may not intuitively know where Figures 2a or 2e are located.

- Line 102 – Could you define more specifically what is the “net attenuation”, i.e. what it is due to?

- Figure 3:

- 1) caption – I usually do not like the terms “statistically significant”. It is too imprecise. You need to define how you measure the statistical significance and attach a value to it. For instance, have you calculated the p-values?

- 2) Figure 3c – I do not know how it will render on printed copies, but the black cross are impossible to see on the electronic copy. I would suggest you use another marker system (e.g. outside line circles?) and/or enlarge the figures.

- Line 140 – “[...] on the combined action of wind speed, wind duration and fetch”. I found the juxtaposition of these terms awkward in this context, because the fetch is not an action but an influencing parameter. Maybe it could be re-phrased as: “[...] the combined influence of wind speed and persistence, and fetch”. It is just a suggestion, I leave it to you to decide.

Also, I do not understand the sentence “[...] it is likely that the fetch is reduced approaching the coast, due to the shadowing effect.” Could you try to re-phrase it better?

- Lines 152-155 – The discussion on the potential cause for the difference between model and altimetry data draws the attention to the spatial averaging, but it seems to me that the difference could be due primarily to the difference in time interval. Am I correct, and if so could you clarify this better?

- Line 154 – Repetition of word “the”.

- Table 2 – I find the use of percentages for the regional comparison a bit misleading, as it depends on the scale of offshore energy flux. For instance the reduction of energy flux in North America (E) and (W) is almost the same but the percentage show a large difference, in the reverse there is a more significant difference between South America (E), with a difference of 3 kW/h, and (W), with a difference of 11.34 kW/h; but the percentages are almost the same.

Which is the most relevant when looking at the energy flux reduction, the absolute difference or the percentage?

- Figure 7b – the y-axis label should read “ratio”
- Lines 286-287 – I would have liked to see figure(s) showing the improvement between available data and reprocessed ALES data. Maybe this is presented in previous paper, but it would be good to have a figure supporting the text here.

Reviewer #2 (Remarks to the Author):

Review of the NCOMMS manuscript “Global coastal attenuation of wave height and wave energy observed with radar altimetry” by Passaro et al.

I am afraid I have to reject this manuscript, since not only it doesn't have enough quality and relevance for a Nature publication, but it has a fundamental error in its core.

There are several minor details in the English language, needless to mention, since I will not let the manuscript go through.

Why in the global maps you chose not to grid the altimetry along tracks and instead plot the contours in the along track data (as in figure 1a, for example)? Never seen it. Not correct. You have to grid the altimetry data, averaging it into boxes (1 or 2 dgrs) and then produce the contours. You also fail to explain your averaging technique properly.

In situ buoys is a redundancy (L61). Wave buoys are always in situ instruments. There are several examples across the text revealing, to a good extent, some lack of knowledge of the field. For example on L84 you say that ERA5 is a model, when it is of course a reanalysis. Also the ERA5 wave reanalysis spatial resolution is not 0.5 dgr. And why do you mention on L85-86 ERA-Interim, and reference it with Dee et al. (2011), and do not provide a reference to ERA5? This would be detail at a different level, but I afraid not at this level. In the same line comes what you state on L103-105 in the sentence starting with “Further...”. At this level you do not publish incomplete research. If your algorithm can be improved get back to work and come back when it is perfect.

Another example is when you present the seasonality of the global SWH, and get very surprised with it. There is nothing new there we have all known that for a very long time.

There is no Asian Summer Monsoon (L121). It is called South Asian Monsoon, and it has an impact in the winds across the Arabian Sea, that shift 180 dgs from boreal winter to summer periods.

In several parts of the text you mention good and excellent agreements with published science, but you fail to quantify it. How excellent? How good? How remarkable? (Needless to say that excellent and remarkable should not be used in this context.)

You cannot compute the climatological mean wave energy flux using altimetry SWH and ERA5 reanalysis mean wave periods. This is fundamentally wrong and cannot be done.

Reviewer #3 (Remarks to the Author):

Page 1. Lines 23-24.

The significant wave height approaches the zero-moment wave height, H_{m0} for sea states with narrow banded spectrum. For these sea states, $H_{m0} = 4.004 (m_0)^{0.5}$ and m_0 can be approximated by the variance of the vertical free surface displacement, i.e., the square of the standard deviation of the free surface vertical displacement. The wave height distribution in a sea state is usually Rayleigh and it is not the same as the free surface vertical displacement that is usually Gaussian.

Page 8, lines 122-123.

Please indicate that figure 3b is the amplitude of the annual cycle in the coastal regions and that figure 3c is the difference between the offshore and the coastal estimate of the amplitude (as indicated in figure 3 caption).

Page 9, figure 4 caption. Please correct "ration"

Page 9, line 147.

The wave energy flux should be obtained through the integration of the product of the wave directional spectrum, by the group celerity, C_g . As the Satellite do not provides a good information about wave frequency and directionality, and nothing about water depth, please explain how the wave energy flux has been obtained. Equation (6) in page 18 is valid for deep water waves, and the wave period is taken from the ERA5 reanalysis not from the satellite information.

Page 12, figure 7.b. Please correct "ration"

Page 12. General comment.

The paper of Losada et al. (2010) presented a new methodology to assess the wave energy resources near the coasts and was applied to the development of the Spanish Atlas of Wave Energy Resource. Satellite wave data and deep water and shallow water wave buoys were employed to calibrate and validate the proposed methodology. The SWAN wave model was used for the propagation of offshore wave climate to the coast (up to 30 m water depth) with a mesh resolution in the hundred m.

In the paper of Pérez et al. (2017) The Global Ocean Waves Two database (GOW2) was presented. GOW2 is a wave hindcast based on WaveWatch III (WWIII) version 4.18 that divides the World Ocean in four regular meshes: the global mesh (0.5° by 0.5°), two regional meshes that cover the Arctic and Antarctic areas (0.25° latitude by 0.5° longitude) and the coastal mesh (0.25° by 0.25°). This coastal mesh includes all the points with water depth below 200 m and a surrounding area of 1.5°. These data bases have been calibrated and validated using all available instrumental data (satellite and wave buoys).

Page 17. Validation.

Validation has been carried out comparing the amplitude of SWH annual cycle with the one of the data obtained by the nearby buoys. This fact could explain the differences in the mean SWH amplitude for some locations.

References. The reference of Passaro et al. (2019) is not cited in the text

Cited bibliography.

Inigo J .Losada, Fernando J. Mendez, César Vidal, Paula Camus, Cristina Izaguirre (2010). Spatial and temporal variability of nearshore wave energy resources along Spain: methodology and results. OCEANS'10 MTS/IEEE Seattle.

Pérez, J., Menéndez, M. and Losada, I.J. (2017). GOW2: A global wave hindcast for coastal applications. Coastal Engineering 124:1-11. DOI: 10.1016/j.coastaleng.2017.03.005.

Reviewer #4 (Remarks to the Author):

Comments on the paper:

“Global coastal attenuation of wave height and wave energy observed with radar altimetry”

by M.Passaro, C.Schwatze, D.Dettmering, F.Seitz

This paper describes the methodology followed for using the Jason 1-2 Hs data extended till practically the coast to derive local global statistics to be then compared with the corresponding offshore (30 km) values. The statistics are, where possible, compared with coastal buoy data. In the following I list first the general positive aspects, then my general criticism, followed by several detailed comments plus my suggestion to the editor.

Positive aspects

- 1 – it is the first time altimeter Hs data are reported so close to the coast on a global basis,
- 2 – the data cover an extended period, and, within limits to be then specified, in this respect they are significant,
- 3 – the methodology appears reliable (but see also comments and criticism),
- 4 – the global scale of the results is duly remarkable.

Criticism

- 1 – the paper has obviously been written by people familiar with satellite altimetry, but not with oceanography. Nothing (almost) of what they say about waves is wrong, but it is often trivial, to be taken for granted. A suggestion could be to involve a wave oceanographer as a co-author. This is usually a convenient approach when dealing with two connected, but different, subjects.
- 2 – the positive value and interpretation of the results is too generous – more criticism is badly required,
- 3 – there is a lot of physics, not discussed or considered, in the interpretation of the results,
- 4 – some assumptions concerning the just mentioned point 3 are simply wrong,
- 5 – the assumptions about fluxes are often not realistic,
- 6 – contrarily to tile and text, fluxes have not been measured, but derived,
- 7 – the used methodology to derive results very close to coast may imply some bias (that I later specify),
- 8 – the large scale (continental) classification of the results is of no practical use.

Detailed comments (also about trivial details) following the line numbers

- 3 – “computationally very expensive”
- 5 – “coastal data ... flagged as unreliable”. It depends on the distance from the coast. Being in the abstract, the implied distance is not obvious.
- 14-15 – obviously the energy fluxes have not been measured
- 33 – “nested into high resolution ..”. Not correct. Better “a nested high resolution model needs to be used in order ..”
- 42 – 10 resolution – I believe this is the resolution of the summary results, not of the original model data. These last years the global resolution of the ECMWF wave model results is 14 km. Also about ERA-5, Jean Bidlot has run a relatively high resolution global wave model for the whole period of ERA5.
- 51 – “brings” ♦ “has brought”
- 72 – better to specify at the beginning of the sentence if “values” refer to Hs or differences
- Figure 1 – b) I suggest to explain better the meaning of the thickness of the stripe bordering the global coasts. I assume it is so thick to make it evident, In this respect the single information in c) is hardly visible, except for a large enlargement on the screen
- 76 – there are also blue points on Figure 1c – this implies the coastal values are higher. This should be commented about and explained.
- Table 1 – no point in specifying the double digit of the fraction of percent
- 85 – ERA5 4 to 7 decades. Originally 4 then extended. Better to be precise.
- 88 – sheltering is not due to dissipation – the overall is not difficult, but not to be summarized in a short sentence. For instance, what about fetch limited conditions?
- 90 – “the effect can occasionally be seen”
- 91 – “can be seen – only enlarging the figure
- 97-98 – “in figure 2d” – I have doubt about this area for limited depth effects. If I remember correctly this

area of the Ireland sea is deep till the coast.

Figure 2 – once printed, the details will be hardly visible. The single panels could be larger. In any case, granted the geographical coordinates, it would be nice to mention which area of the world the reader is looking at. If I am not mistaken, in order: West Canada, top of Scotland, Sri Lanka, Ireland, Maldives, Terranova -- In this figure I am puzzled by one detail: all the wave directions go practically West to East. Ok, this is the general flow, but I find strange that the mean values are so uniform, also at practically the equator.

108 and following till 121 – I do not see the purpose of this part of the text. What reported is either trivial or common knowledge. On the contrary it is presented as a result. No doubt, but the maximum the authors can say is that the results are consistent with what already known. In a way this should be taken for granted. What else? The same is true for the following paragraph.

127-132 – the comparison between the east side and west one of an ocean is not so simple. The whole interpretation is by far too simplistic.

139-140 – this kind of explanation would hardly be suitable for a first year course. There is no need of quoting Hwang or any other paper for this.

140-145 – all this discussion avoids the point if the wind is blowing offshore or onshore. If towards the coast, there no difference between swell and wind sea attenuation.

148-149 – there is nothing to confirm. Given that power depends on H^2T , the predominance of H is obvious

152-155 – I wonder about this statement. In a random sampling (in this case random with respect to the sea conditions) the mean should not depend on the sampling frequency. What varies are the confidence limits of the result. The matter would be different if looking for extremes.

Figure 6 – a few things. Here we are zooming on an area to see the details. However, the vertical scale spans 10 degrees, 1111.11 km. If the details the reader should look at are at 30 km or less close to coast, this will correspond to 1/37 of the vertical scale of the figure, or less. This appears by far too little. In any case, if I interpret properly the figure, it looks to me that in the altimeter results (central part of the figure) the blue colour extends offshore more than the background that I interpret as what computed by Hemer et al.

Table 2 – these large scale values are of little help given the enormous variability within each indicated area. Also, as previously, there is no point in specifying the second decimal digit of a percent.

181-182 – this is not a general rule. It depends on the bottom topography, general geometry, et al. Last but not least, if the wind is blowing offshore or onshore.

185-189 – these average conclusions are of little help or use given the spreading of the results, especially for energy flux.

...-197 – see comment at the beginning. Datasets exist with 14 km resolution for the recent few years, slightly more coarse for the full ERA5 period.

217 – here we go back to my comment at the beginning- There are trivial explanations for wave oceanography, and professional ones for altimetry and related analysis. Given some trivial arguments about wave physics, for the benefit of the oceanographic readers (the ones more interested in the results), possibly a little explanation of the ALES algorithm could be useful.

Figure 8 – horizontal units are missing

256 – do I understand correctly that the authors derive one value every 350 m?

261-262 – what else?

following – it appears complicated in the text, but it is fitting a sinusoid plus a trend. I agree that the iterative procedure is nice, but I wonder if it was really required. For instance, given the approximation of the data, which would be the differences with respect to taking a simple fit or even more simply the differences between the sequential max and min minimal values?

287 – “up to about 3 km” – I have a general question – going very close to coast implies probably more outliers, especially in a rough sea (a wider reflection zone). Through the described “cleaning” and selection procedure, this could affect the coastal values, hence statistics.

Independently of the paper I take this chance for a general question about which I ask an explanation by the authors. I have been using altimeter data for a long while of course, typical interval 7 km at 1 s interval. I understand each value is the average of 20 samples at 20 Hz uniformly distributed along the 7 km. Once these 20 data are averaged into a single value at a specified position, where is this position located along the 7 km? While I had this question for a while, I was wondering about it reading in this paper about going officially till 3 km off the coast. Thank you.

289-290 – this is connected to a previous comment. There is no doubt much lower number of altimeter data at buoy location will lead to less accurate results, but it should not affect the mean value in itself. This granted that the time of the day is not significant.

296 – at this level of sophistication 30 km can be relevant, especially close to coast where conditions are likely to change at shorter space interval. Indeed, the authors choose a max 15 km difference. Given the distance between adjacent satellite tracks (200 km or more), how general can the results be considered? This should be discussed.

300 – “excellent agreement”. I never appreciate the qualitative appreciation, especially self-ones by authors. Results should be reported in a quantitative way, defining them useful or not for a specific purpose or another. In the present case, going back to what said above about Figure 6. I would have appreciated, and it would have been much more realistic, to zoom at the scale we are dealing with, 30 km or so, e.g. on the central part of Figure 6, and compare on this suitable zone the Hemer and altimeter results. From the figure I derive they are not so coincident.

313-316 – is this true also in areas (most of the ocean coasts) with mixed wind sea - swell?

326 – “waves ..., generated locally”. The argument is not posed properly. Are we talking about offshore or inshore blowing wind? If towards the coast, waves are continuously generated till the coast. If from the coast, we go back to the point I mentioned above.

331 – same comment as above about Figure 6.

Conclusions

Basically the valuable part of the work done by the authors is the extension closer to coast of the results derivable from the Jason 1-2 Hs data. This is remarkable. There is some validation, but in my opinion not done in an objective way (e.g. Figure 6, if I have interpreted properly its significance). The so-called results obtained in the open ocean are obviously a repetition of what is presently common knowledge. There is a lack of proper discussion and analysis of the relationship between wind and generated waves in the various possible situations. There is no discussion of the possibility of inshore and offshore blowing wind.

My final suggestion is to reject the paper encouraging a resubmission following the above suggestions and criticism. However, I strongly recommend that the authors look for the help of an oceanographer to deal in a more apt way with the physics of the marine part. If following this suggestion, for the benefit of the authors (but it may be already the approach by Nature Communications) I suggest to maintain the same editor and reviewers.

Luigi Cavaleri

First of all we would like to thank all the reviewers for their analysis and suggestions. We believe the study has gained a lot from the reviewing process and we hope our answers and changes match the reviewers' expectations.

Our point-by-point answer is provided below. The text of the answer to the reviewers in this document is in italic and highlighted. The corresponding changes in the manuscript are in red.

Reviewers' comments:

Reviewer #1 (Remarks to the Author):

General comments:

The ALES algorithm is applied to characterise the ocean waves attenuation in the coastal zone, from altimetry data (Jason-1 and Jason-2 missions). The paper gives a useful observation of the significant wave height and energy flux at a global scale both offshore and nearshore. Such a coverage can only be achieved by satellite altimetry. The application of the ALES retracker algorithm, detailed in previous papers from the same lead author, gives more reliable data close to the coast than previous altimetry data. Therefore this presents a potentially wide interest for a range of applications, such as coastal erosion/protection, model calibration, wave energy site development, or coastal maritime traffic, which justifies its publication.

The paper will influence the wave resource analysis because it solves one of the major drawback of altimetry data (unreliability of altimetry data near the coast), and it applies the method to a worldwide screening of coastal wave energy.

Overall it is an good application of the ALES retracker algorithm to estimate the global coastal wave height and energy flux.

Detailed comments:

- The paper is articulated around two different parts presenting the methods and the results. The paper reads well, although language and clarity could be improved in some cases. The section on methods better written and structured, but I was surprised to see that section after the results section. Not sure why the manuscript was structured that way, as it seems to me more logical to present the methods before the results and conclusions.

Is it possible to restructure the paper by putting section 4 before sections 2 and 3?

We followed the general structure of the papers sent to Nature Communications, where Methods are given in a subsequent section that follows the conclusions. We agree that this is different from the structure of most of the other peer-reviewed journals in our field. Still, given the requests of the journal, we would not change the order of the sections unless it is requested by the Editor

- Line 21 – “more in general” should better read “more generally”

Done

- Line 91-93 – I found the illustrative example of the Azores and Kerguelan archipelagos a bit weak, as it is not easy to spot it in Figure 1a, due to the scale and resolution of the figure. Could it be a more obvious example (e.g. Cape Horn?) ?

We agree with the Reviewer that the examples are hardly visible on a global scale. Nevertheless, the sheltering effect seen at the latitude of Cape Horn and discernible from a global scale is actually attributable to the Patagonian peninsula and therefore does not fit the text, which refers to islands. We have decided to modify the text making the citation of Ponce de Leon et al. (2005) more explicit and restricting the example to the Azores:

“For example, Ponce de Leon and Guedes Soares, 2005, report the broad scale effects of sheltering from the Azores Archipelago on the Atlantic wave climate.”

- Figures 2:

- 1) axis and colorbar labels are too small. Could you increase their font size or enlarge the figures?
- 2) although it can be deduced from the longitude and latitude coordinates (and for most graphs the coastlines configuration), it would be good to name the locations of each plot, either in a plot title, or in the caption. For instance, readers may not intuitively know where Figures 2a or 2e are located.

Both points shall be now taken into account in the new Figure and its caption

- Line 102 – Could you define more specifically what is the “net attenuation”, i.e. what it is due to?

By “net attenuation” we meant the combined effect of the phenomena already described few lines before in the previous paragraph (shading, diffraction, seabed friction). We agree that the term “net attenuation” might be misleading and we have therefore rephrased the paragraph in the following form:

“While our observations show that on average the attenuation of SWH from offshore up to 3 km from the coast is prevalent, this does not exclude that locally the average SWH can increase in the last 3 km. This limitation might be overcome in the next years, when Delay-Doppler altimeters on repeated tracks will have acquired time series that are long enough to observe a mean behaviour. These altimeters are characterized by a better signal-to-noise ratio and along-track resolution, which could enable to fill the remaining coastal gap.”

- Figure 3:

- 1) caption – I usually do not like the terms “statistically significant”. It is too imprecise. You need to define how you measure the statistical significance and attach a value to it. For instance, have you calculated the p-values?

We added in the caption information on the adopted definition of significance and the reference to the methods for the computation of the uncertainty: “The point is marked as significant if the absolute value of the difference between the coastal and the offshore amplitude is higher than its uncertainty. Uncertainty are computed as described in section 4.2”

2) Figure 3c – I do not know how it will render on printed copies, but the black cross are impossible to see on the electronic copy. I would suggest you use another marker system (e.g. outside line circles?) and/or enlarge the figures.

We followed the suggestion and used outside line circles

- Line 140 – “[...] on the combined action of wind speed, wind duration and fetch”. I found the juxtaposition of these terms awkward in this context, because the fetch is not an action but an influencing parameter. Maybe it could be re-phrased as: “[...] the combined influence of wind speed and persistence, and fetch”. It is just a suggestion, I leave it to you to decide.

Also, I do not understand the sentence “[...] it is likely that the fetch is reduced approaching the coast, due to the shadowing effect.” Could you try to re-phrase it better?

Based on the suggestions of the other reviewers, the discussion on the wind fetch and its influence on the wave parameters considered in this study has now been modified and the former Figure 4 has been removed.

- Lines 152-155 – The discussion on the potential cause for the difference between model and altimetry data draws the attention to the spatial averaging, but it seems to me that the difference could be due primarily to the difference in time interval. Am I correct, and if so could you clarify this better?

We thank the reviewer to underline the remaining differences between model and altimetry data offshore, considering the average wave energy flux. In the updated figure, we show that this difference is not present anymore and therefore the related discussion of the former lines 152-155 has been removed. The difference was due to the fact that we were computing the average wave energy flux based on the mean wave period, instead of using instantaneous values for each point in the time series. This has now been corrected, as explained also in section “Computation of the average wave energy flux”, in the methodology.

- Line 154 – Repetition of word “the”.

Corrected

- Table 2 – I find the use of percentages for the regional comparison a bit misleading, as it depends on the scale of offshore energy flux. For instance the reduction of energy flux in North America (E) and (W) is almost the same but the percentage show a large difference, in the reverse there is a more significant difference between South America (E), with a difference of 3 kW/h, and (W), with a difference of 11.34 kW/h; but the percentages are almost the same.

Which is the most relevant when looking at the energy flux reduction, the absolute difference or the percentage?

We agree with the reviewer that for users interested in the results, the absolute values in terms of energy flux are more important than the change in percentage. Moreover, the relative change at a global scale is already highlighted in another figure. Therefore, in the table the absolute difference is reported, instead of the percentage.

- Figure 7b – the y-axis label should read “ratio”

Corrected

- Lines 286-287 – I would have liked to see figure(s) showing the improvement between available data and reprocessed ALES data. Maybe this is presented in previous paper, but it would be good to have a figure supporting the text here.

We fully understand the reviewer’s request and we think this can improve the paper. Therefore, we have added a new figure in section 4.3 showing the difference in correlation between buoy and altimetry dataset, comparing results with the standard altimetry dataset against the ALES-reprocessed one used in this study.

Reviewer #2 (Remarks to the Author):

Review of the NCOMMS manuscript “Global coastal attenuation of wave height and wave energy observed with radar altimetry” by Passaro et al.

I am afraid I have to reject this manuscript, since not only it doesn't have enough quality and relevance for a Nature publication, but it has a fundamental error in its core.

We hope that the quality and relevance of our manuscript has been increased and clarified in this review and according to the answers to all reviewers' comments. The reviewer disagrees with our along-track approach and the computation of the mean wave energy flux using altimetry SWH and ERA5 reanalysis mean wave periods. Despite the lack of further explanations, concerning the first point we have tried to clarify why we disagree and we report in the following lines that this methodology has been already used in relevant previous studies. We have also increased the validation to show the validity of our approach. Concerning the second point, we understand the concern of the reviewer and we have therefore corrected our approach in this review: the wave energy flux is now computed for every point of the altimetry track using the instantaneous estimation of SWH and the closest value in time of the relevant wave period from the ERA5 reanalysis (which is distributed with a 3-hour time resolution). This approach has improved the results and the comparison with modelled data.

There are several minor details in the English language, needless to mention, since I will not let the manuscript go through.

In the new submission, an English native speaker is among the authors and has double checked the use of English

Why in the global maps you chose not to grid the altimetry along tracks and instead plot the contours in the along track data (as in figure 1a, for example)? Never seen it. Not correct. You have to grid the altimetry data, averaging it into boxes (1 or 2 dgrs) and then produce the contours. You also fail to explain your averaging technique properly.

It is not clarified in the review which would be the fundamental error in the core of this study according to the reviewer. From the sentence above, we assume the reviewer thinks that the fact that altimetry time series are created along the track instead of gridding is a fundamental error. We do not think that this opinion is justified. Surely we agree that in the standard literature, as reported in the Introduction of this paper, the data are averaged into boxes (1 or 2 dgrs). The reviewer will agree with us that for the specific purpose of this paper, which is to analyse differences within 30 km of the coast, this methodology would not make sense, since a 1-degree grid has a spacing of several tens of kms.

We understand the concern that there is no reason not to grid when showing results at a global, very large scale. The objective of the global figures is not to reveal new insights (as in the case of the coastal analysis), but to show the consistency of this new dataset and of the along-track analysis with the knowledge acquired up to now with gridded data.

Figure 9 demonstrates, by comparison with buoys, that an along-track analysis based on repeated tracks is able to create time series from which the parameters of interest for this study (mean SWH and seasonal cycles) can be correctly estimated. We have never said that the along-track analysis can be used for all purposes. In fact, we do not claim to assess trends in SWH climate, which would indeed have an uncertainty that is way too high for a statistical analysis.

The reviewer states “Never seen it. Not correct”. In general, we do not believe that a scientific method that is never seen, is automatically not correct. We also do not see any proof or reason of the incorrectness of our methodology in the reviewer’s statement. Finally, if an along-track analysis has not been performed with SWH along-track data yet, this has been and is being performed for mean sea level and annual cycle of the sea level at coastal scales (examples: Gómez-Enri et al. (2018), Marti et al. (2019), Passaro et al. (2015), Passaro et al., 2016). In those cases, in which the subject of the analysis is sea level, results are validated against tide gauges. In our case, since we deal with SWH, results are validated against buoys.

Gómez-Enri J., González C.J., Passaro M., Vignudelli S., Álvarez O., Cipollini P., Mañanes R., Bruno M., López-Carmona M.P., Izquierdo A.: Wind-induced cross-strait sea level variability in the Strait of Gibraltar from coastal altimetry and in-situ measurements. *Remote Sensing of Environment*, 221, <https://doi.org/10.1016/j.rse.2018.11.042>, 2018

Marti F., Cazenave A., Birol F., Passaro M., Leger F., Nino F., Almar R. Benveniste J., Legeais J.F.: Altimetry-based sea level trends along the coast of Western Africa. *Advances in Space Research*, 10.1016/j.asr.2019.05.033, 2019

Passaro M., Cipollini P., Benveniste J.: Annual sea level variability of the coastal ocean: The Baltic Sea-North Sea transition zone. *Journal of Geophysical Research* 120(4): 3061-3078, 10.1002/2014JC010510, 2015

Passaro M., Dinardo S., Quartly G.D., Snaith H.M., Benveniste J., Cipollini P., Lucas B.: Cross-calibrating ALES Envisat and CryoSat-2 Delay–Doppler: a coastal altimetry study in the Indonesian Seas. *Advances in Space Research*, 58(3), 289-303, 10.1016/j.asr.2016.04.011, 2016

In situ buoys is a redundancy (L61). Wave buoys are always in situ instruments.

Corrected

There are several examples across the text revealing, to a good extent, some lack of knowledge of the field.

In the new submission, we benefit from the collaboration with Dr. Mark Hemer, whose primary research interest focus on the interaction of waves with other oceanographic drivers in the nearshore and coastal zone. This collaboration between remote sensing experts and wave experts shall enable to fill the gaps of the previous version.

For example on L84 you say that ERA5 is a model, when it is of course a reanalysis.

L84 of the previous version did not say that ERA5 is a model, but defined ERA5 as a “reanalysis model”. The term “reanalysis” appeared further 8 times in the manuscript. Nevertheless we agree that ERA5 shall be referred as “reanalysis” and not as “reanalysis model”. Following the definition reported in [1]: “Reanalyses provide a numerical description of the recent climate by combining models with observations.”. This is now adopted throughout the manuscript.

[1] <https://www.ecmwf.int/en/newsletter/147/news/era5-reanalysis-production>

Also the ERA5 wave reanalysis spatial resolution is not 0.5 dgr.

We attach here the table available from the link that describes ERA5 wave reanalysis data, from which data are downloaded, as reported in the Acknowledgements. According to this table, if we are wrong, then also the official description is wrong. Nevertheless we acknowledge that the original resolution of the model used for the reanalysis may be higher [1], therefore we correct the sentence as follows: “...provided by the ERA5 reanalysis on a 0.5 x0.5 grid...”

We also report the text of a personal communication from Jean Bidlot, coauthor of the ERA5:

“...part of the confusion might come from the fact that ERA5 is NOT using the same resolution as our operational forecast system...Our operational forecast wave products are produced on a native grid with a resolution ~14km, which are made available to users on 0.1x0.1 or 0.125x0.125 degree grids whereas, ERA4 is produced on a native grid with a resolution ~40km, made available via the CDS on 0.5x0.5 degree grid”

[1] <https://www.ecmwf.int/en/about/media-centre/news/2019/upgrade-boost-quality-ocean-wave-forecasts>

DATA DESCRIPTION	
Data type	Gridded
Horizontal coverage	Global
Horizontal resolution	Reanalysis: 0.25°x0.25° (atmosphere), 0.5°x0.5° (ocean waves) Mean, spread and members: 0.5°x0.5° (atmosphere), 1°x1° (ocean waves)
Temporal coverage	1979 to present
Temporal resolution	Monthly
File format	GRIB
Update frequency	Monthly

And why do you mention on L85-86 ERA-Interim, and reference it with Dee et al. (2011), and do not provide a reference to ERA5? This would be detail at a different level, but I afraid not at this level.

We trust that the reviewer is fully aware that no peer-reviewed publication about ERA5 has been available at the time of the first submission. “At this level”, as the reviewer correctly notices, only peer-reviewed paper shall be cited. At the time of revision, we got to know that the following paper concerning ERA5 has been accepted. This is now adopted instead of Dee et al. (2011):

Hersbach et al. 2020 (<https://rmets.onlinelibrary.wiley.com/doi/abs/10.1002/qj.3803>) Please note that this article was first published on the 17 May 2020 and therefore a previous citation could not have been possible.

In the same line comes what you state on L103-105 in the sentence starting with "Further...". At this level you do not publish incomplete research. If your algorithm can be improved get back to work and come back when it is perfect.

Firstly we would like to mention that no algorithm is perfect and perfection does not belong to science. Each algorithm and technique has its limitation and it is essential to know these limits. There are countless examples for this. To remain in the context of satellite altimetry, we can refer to the fact that, although the current algorithms used to derive sea level have significant problems in the signal-to-noise ratio for scales below ~100-50 km (depending on the mission and the region), this has not prevented numerous studies aware of this limitation (ex.:Morrow et al., 2017). This is even true although we perfectly know that those algorithms can be (and are being) improved (Quartly et al., 2019).

Secondly, the 3 km limitation cannot be fully overcome with Low Resolution Mode altimeters, since this is essentially the footprint defined by the leading edge of the pulse limited radar footprint itself.. The statement therefore has the aim to pave the way for the exploitation of Delay-Doppler altimeters in the next years to fill the remaining coastal gap. We have rephrased the paragraph in this way:

"This limitation might be overcome in the next years, when Delay-Doppler altimeters on repeated tracks will have acquired time series that are long enough to observe a mean behaviour. These altimeters are characterized by a better signal-to-noise ratio and along-track resolution, which could enable to fill the remaining coastal gap."

Rosemary Morrow, Alice Carret, Florence Birol, Fernando Nino, Guillaume Valladeau and Francois Boy, "Observability of fine-scale ocean dynamics in the northwestern Mediterranean Sea." Ocean Science, vol. 13, no. 1, 2017, p. 13.

Quartly G.D., Smith W., Passaro M.: Removing Intra-1-Hz Covariant Error to Improve Altimetric Profiles of σ° and Sea Surface Height. IEEE Transactions on Geoscience and Remote Sensing, 1-12, 10.1109/tgrs.2018.2886998, 2019

Another example is when you present the seasonality of the global SWH, and get very surprised with it. There is nothing new there we have all known that for a very long time.

As suggested also by another reviewer, the discussion has been modified throughout the paper in order to avoid stressing too much elements of the SWH variability that are already very well known. We trust this can be appreciated in this version. The presence of the global offshore plots are not aimed at presenting new findings, but at providing a comparison with the differences found with the coastal values and at proving that the dataset we are using, which reveals new insights for the coastal zone, is fully capable of reproducing the known offshore characteristics as well.

There is no Asian Summer Monsoon (L121). It is called South Asian Monsoon, and it has an impact in the winds across the Arabian Sea, that shift 180 dgs from boreal winter to summer periods.

We adopt the definition "South Asian Summer Monsoon" (the "South" is now added to the line the reviewer mentions). This is fully in line with a very wide literature, for example:

[1] Turner, A. G., & Annamalai, H. (2012). Climate change and the South Asian summer monsoon. Nature Climate Change, 2(8), 587-595.

[2] Bollasina, M. A., Ming, Y., & Ramaswamy, V. (2011). Anthropogenic aerosols and the weakening of the South Asian summer monsoon. *science*, 334(6055), 502-505.

[3] Singh, D., Tsiang, M., Rajaratnam, B., & Diffenbaugh, N. S. (2014). Observed changes in extreme wet and dry spells during the South Asian summer monsoon season. *Nature Climate Change*, 4(6), 456-461.

In several parts of the text you mention good and excellent agreements with published science, but you fail to quantify it. How excellent? How good? How remarkable? (Needless to say that excellent and remarkable should not be used in this context.)

In this revision, such qualitative speculations are removed and substituted when needed by further validation

You cannot compute the climatological mean wave energy flux using altimetry SWH and ERA5 reanalysis mean wave periods. This is fundamentally wrong and cannot be done.

The wave energy flux is now computed for every point of the altimetry track using the instantaneous estimation of SWH and the closest value in time of the relevant wave period from the ERA5 reanalysis (which is distributed with a 3-hour time resolution). A similar approach (i.e. coupling wave periods from the model Wavewatch III with SWH from altimetry) was used in previous studies such as Young et al., 2013. Finally, the validation of our approach against a fully modelled approach as in Hemer et al. is now provided in Figure 5b

Young, I. R., Babanin, A. V., & Zieger, S. (2013). The decay rate of ocean swell observed by altimeter. *Journal of physical oceanography*, 43(11), 2322-2333

Reviewer #3 (Remarks to the Author):

Page 1. Lines 23-24.

The significant wave height approaches the zero-moment wave height, H_{m0} for sea states with narrow banded spectrum. For these sea states, $H_{m0} = 4.004 (m_0)^{0.5}$ and m_0 can be approximated by the variance of the vertical free surface displacement, i.e., the square of the standard deviation of the free surface vertical displacement. The wave height distribution in a sea state is usually Rayleigh and it is not the same as the free surface vertical displacement that is usually Gaussian.

The reviewer is using another definition of the significant wave height (as in Holthuijsen, 2007) and it is making us aware of the issue of the wave height distribution vs surface vertical displacement. On the other side, given the focus on satellite altimetry, we have to stick to the definitions that were used when modelling the pulse-limited altimetry footprint on which the physical model currently in use to derive SWH for these altimeters is based (Chelton, 2001). To take in consideration both of these points, we reword the statement as "The significant wave height (SWH), defined as four times the standard deviation (std) of the surface elevation [Arduin et al., 2019], is an integral parameter that is extensively used as reference to quantify both extremes and mean sea states."

Holthuijsen, L. H., & Booij, N. (2007). Experimental wave breaking in SWAN. In Coastal Engineering 2006: (In 5 Volumes) (pp. 392-402).

D. B. Chelton, J. C. Ries, B. J Haines, L. Lueng Fu, and P. S Callahan. Satellite altimetry. In L. Lueng Fu and A. Cazenave, editors, Satellite Altimetry And Earth Sciences: A Handbook Of Techniques And Applications, volume 69, pages 1{131. Academic Press, 2001.

Page 8, lines 122-123.

Please indicate that figure 3b is the amplitude of the annual cycle in the coastal regions and that figure 3c is the difference between the offshore and the coastal estimate of the amplitude (as indicated in figure 3 caption).

This is now corrected

Page 9, figure 4 caption. Please correct "ration"

The former figure 4 has been removed following the comments of the other reviewers.

Page 9, line 147.

The wave energy flux should be obtained through the integration of the product of the wave directional spectrum, by the group celerity, C_g . As the Satellite do not provides a good information about wave frequency and directionality, and nothing about water depth, please explain how the wave energy flux has been obtained. Equation (6) in page 18 is valid for deep water waves, and the wave period is taken from the ERA5 reanalysis not from the satellite information.

The wave period derived from altimeters is viewed by wave community with skepticism, so we think making use of ERA5 data is probably a better path forward. Indeed, this approach is in-line with that used by Young et al. (2013), who made use of a wavewatch-iii simulation to provide direction and wavenumber (period).

Following the request of explanation considering the adoption of the deep water equation, we have performed an analysis which we report here for the benefit of the reviewer. The adoption of equations adapted for intermediate and shallow water requires knowledge of the wave lengths and the bathymetry. In order to understand the error that we are committing in using the deep water assumption, we use the bathymetry from GEBCO2020 and the wave period from ERA5 to derive an estimation of the wavelength following the parameterisation of Guo, J. (2002).

Following the linear wave theory, the wave energy flux is $P=Ec$, where c is the group velocity and $E=\rho \cdot g \cdot SWHH^2/16$. Using the parameterized mean wavelength, we recompute the average wave energy flux considering the difference in group wave celerity differentiating between deep, intermediate and shallow water, following the equations in Reeve et al. (2012).

The Figure below shows the 98th percentile of the differences between the computation of the average energy flux using the deep water assumption and the approximate solution considering shallow and intermediate waters. The data are binned every km according to their distance to the coast. We conclude that by extending the deep water assumption in all our domain of study we are committing an error that does not exceed 1.1 kW/m.

Guo, J. (2002) Simple and explicit solution of wave dispersion equation, Coastal Engineering 45, 71–74.

Reeve, D., Chadwick, A. J., Fleming, C. (2012). Coastal Engineering: Processes, Theory and Design Practice (2nd ed) E & FN Spon.

Page 12, figure 7.b. Please correct “ration”

Corrected

Page 12. General comment.

The paper of Losada et al. (2010) presented a new methodology to assess the wave energy resources near the coasts and was applied to the development of the Spanish Atlas of Wave Energy Resource. Satellite wave data and deep water and shallow water wave buoys were employed to calibrate and validate the proposed methodology. The SWAN wave model was used for the propagation of offshore wave climate to the coast (up to 30 m water depth) with a mesh resolution in the hundred m.

In the paper of Pérez et al. (2017) The Global Ocean Waves Two database (GOW2) was presented. GOW2 is a wave hindcast based on WaveWatch III (WWIII) version 4.18 that divides the World Ocean in four regular meshes: the global mesh (0.5° by 0.5°), two regional meshes that cover the Arctic and Antarctic areas (0.25° latitude by 0.5° longitude) and the coastal mesh (0.25° by 0.25°). This coastal mesh includes all the points with water depth below 200 m and a surrounding area of 1.5°. These data bases have been calibrated and validated using all available instrumental data (satellite and wave buoys).

We thank the reviewer for the useful references: we have adopted them as a suitable references for the conclusions.

Page 17. Validation.

Validation has been carried out comparing the amplitude of SWH annual cycle with the one of the data obtained by the nearby buoys. This fact could explain the differences in the mean SWH amplitude for some locations.

References. The reference of Passaro et al. (2019) is not cited in the text

This is simply a misunderstanding because in the previous version the reference Arduin et al. (2019) (with Passaro as co-author) was split in two at the second line because of the change of page. There is no Passaro et al. (2019) and Arduin et al. (2019) is cited in the text

Cited bibliography.

Inigo J .Losada, Fernando J. Mendez, César Vidal, Paula Camus, Cristina Izaguirre (2010). Spatial and temporal variability of nearshore wave energy resources along Spain: methodology and results. OCEANS'10 MTS/IEEE Seattle.

Pérez, J., Menéndez, M. and Losada, I.J. (2017). GOW2: A global wave hindcast for coastal applications. Coastal Engineering 124:1-11. DOI: 10.1016/j.coastaleng.2017.03.005.

Reviewer #4 (Remarks to the Author):

Comments on the paper:

“Global coastal attenuation of wave height and wave energy observed with radar altimetry”
by M.Passaro, C.Schwatze, D.Dettmering, F.Seitz

This paper describes the methodology followed for using the Jason 1-2 Hs data extended till practically the coast to derive local global statistics to be then compared with the corresponding offshore (30 km) values. The statistics are, where possible, compared with coastal buoy data. In the following I list first the general positive aspects, then my general criticism, followed by several detailed comments plus my suggestion to the editor.

Positive aspects

- 1 – it is the first time altimeter Hs data are reported so close to the coast on a global basis,
- 2 – the data cover an extended period, and, within limits to be then specified, in this respect they are significant,
- 3 – the methodology appears reliable (but see also comments and criticism),
- 4 – the global scale of the results is duly remarkable.

We provide here a general review of the answers to the main criticisms of the reviewer, while a detailed answer is provided later on following the reviewer's points.

Criticism

- 1 – the paper has obviously been written by people familiar with satellite altimetry, but not with oceanography. Nothing (almost) of what they say about waves is wrong, but it is often trivial, to be taken for granted. A suggestion could be to involve a wave oceanographer as a co-author. This is usually a convenient approach when dealing with two connected, but different, subjects.

We have welcomed the suggestion of the reviewer and we have significantly reviewed the article with Dr Mark Hemer, who is now listed as second author.

- 2 – the positive value and interpretation of the results is too generous – more criticism is badly required,

We have taken care of avoiding any personal interpretation of the results in this new version. We have produced quantitative validation in two new figures (Figure 5b and Figure 8).

- 3 – there is a lot of physics, not discussed or considered, in the interpretation of the results,

- 4 – some assumptions concerning the just mentioned point 3 are simply wrong,

Concerning points 3 and 4: We have completely reviewed Figure 2 in order to select relevant case studies which are then discussed in the text in terms of their physics, to aid the general interpretation of the results. Trivial and common knowledge has been removed throughout the text. We surely agree with the reviewer that our paper focuses more on results and availability of information than on physics.

- 5 – the assumptions about fluxes are often not realistic,

The wave energy flux is now computed for every point of the altimetry track using the instantaneous estimation of SWH and the closest value in time of the relevant wave period from the ERA5 reanalysis (which is distributed with a 3-hour time resolution). A similar approach (i.e. coupling wave periods from the model Wavewatch III with SWH from altimetry) was used in previous studies such as Young et al., 2013. Finally, the validation of our approach against a fully modelled approach as in Hemer et al. is now provided in Figure 5b.

The error due to the use of the deep water assumption has been quantified and reported in the answer to Reviewer #3, referred as "Page 9, line 147."

Young, I. R., Babanin, A. V., & Zieger, S. (2013). The decay rate of ocean swell observed by altimeter. Journal of physical oceanography, 43(11), 2322-2333

6 – contrarily to tile and text, fluxes have not been measured, but derived,

This is now corrected.

7 – the used methodology to derive results very close to coast may imply some bias (that I later specify),

This point is clarified alongside the specification of the reviewer, later on.

8 – the large scale (continental) classification of the results is of no practical use.

While we agree with the reviewer that for practical use of many users the local value is more useful than the large scale summary, we would like to keep it as a tool for resuming our main findings and especially as comparison with other relevant literature in the field of wave energy flux analysis, which will highlight the steps forward brought by this study.

Detailed comments (also about trivial details) following the line numbers

3 – "computationally very expensive"

Corrected

5 – "coastal data ... flagged as unreliable". It depends on the distance from the coast. Being in the abstract, the implied distance is not obvious.

We clarified: "...typically flagged as unreliable within 30 km of the coast."

14-15 – obviously the energy fluxes have not been measured

We have changed the title into "Global coastal attenuation of wind-waves observed with radar altimetry"

33 – "nested into high resolution ..". Not correct. Better "a nested high resolution model needs to be used in order .."

Rephrased as suggested

42 – 1o resolution – I believe this is the resolution of the summary results, not of the original model data. These last years the global resolution of the ECMWF wave model results is 14 km. Also about ERA-5, Jean Bidlot has run a relatively high resolution global wave model for the whole period of ERA5.

The reviewer is right and, in order to avoid confusion and imprecision, we changed the statement in “...provided by the ERA5 reanalysis on a 0.5 x0.5 grid...”

We also report the text of a personal communication from Jean Bidlot, coauthor of the ERA5:

“...part of the confusion might come from the fact that ERA5 is NOT using the same resolution as our operational forecast system...Our operational forecast wave products are produced on a native grid with a resolution ~14km, which are made available to users on 0.1x0.1 or 0.125x0.125 degree gridswhereas, ERA4 is produced on a native grid with a resolution ~40km, made available via the CDS on 0.5x0.5 degree grid”

51 – “brings” ◇ “has brought”

Corrected

72 – better to specify at the beginning of the sentence if “values” refer to Hs or differences

“mean SWHs” substitutes now “values”

Figure 1 – b) I suggest to explain better the meaning of the thickness of the stripe bordering the global coasts. I assume it is so thick to make it evident, In this respect the single information in c) is hardly visible, except for a large enlargement on the screen

To improve the visibility of panel c), the points are now overlayed on the coast as in the other panels. What we are plotting is not a stripe: each point represents one altimetry track. To clarify this, we added the following text: “For each altimetry track, the circles of panel b and c are centred on the coordinates of the coastal location being compared.”

76 – there are also blue points on Figure 1c – this implies the coastal values are higher. This should be commented about and explained.

In order to comment and explain possible reasons for the “blue points”, we have added an example in Figure 2d. The following comment is added: “Few locations in our dataset show no change of mean SWH between offshore and the coast, or a slight increase. One example is provided in Figure 2d, located in Eastern Australia in the region of the Great Barrier Reef. Here, the SWH attenuation caused by the reef (visible by the bathymetric contour at -20 m depth) is counteracted by additional growth on the landward side of the reef. Shallow-water interactions may also drive an increase in SWH, whether via shoaling or convergence of wave energy, as for example via refraction around headlands.”

Table 1 – no point in specifying the double digit of the fraction of percent

Following the comment of another reviewer, this table now reports the absolute differences instead of the percentage

85 – ERA5 4 to 7 decades. Originally 4 then extended. Better to be precise.

We agree with the reviewer, nevertheless the new availability of a peer-reviewed paper describing ERA5 made the previous specification concerning the difference between ERA-Interim and ERA5 no more needed and this sentence is therefore removed in the reviewer manuscript.

88 – sheltering is not due to dissipation – the overall is not difficult, but not to be summarized in a short sentence. For instance, what about fetch limited conditions?

Figure 2 and the related discussion have been fully reviewed in this version, also considering this comment. We now show a case study in which we resolve a sheltering effect from an island (panel a), a case study representative for the depth induced dissipation of wave energy (panel b) and a case study with local wind-generated growth in fetch-limited conditions (panel c).

90 – “the effect can occasionally be seen”

91 – “can be seen – only enlarging the figure

The panels in Figure 2d are now enlarged compared to the previous version

97-98 – “in figure 2d” – I have doubt about this area for limited depth effects. If I remember correctly this area of the Ireland sea is deep till the coast.

Figure 2 has been reevaluated due to an inconsistency in the wave directions, which the reviewer helped to spot. In the new Figure, the example concerning limited depth effects is taken in South Australia. In order to clarify the depth issue, we also highlight the “-20” m depth contour.

Figure 2 – once printed, the details will be hardly visible. The single panels could be larger. In any case, granted the geographical coordinates, it would be nice to mention which area of the world the reader is looking at. If I am not mistaken, in order: West Canada, top of Scotland, Sri Lanka, Ireland, Maldives, Terranova

Panels have been enlarged and the area of the world for each subplot is now reported in the caption.

-- In this figure I am puzzled by one detail: all the wave directions go practically West to East. Ok, this is the general flow, but I find strange that the mean values are so uniform, also at practically the equator.

We thank the reviewer for spotting this inconsistency. Indeed, the arrows representing the mean wave direction were plotted wrongly due the use of radians in a GMT plot function that accepts degrees. This is amended in the new figure.

108 and following till 121 – I do not see the purpose of this part of the text. What reported is either trivial or common knowledge. On the contrary it is presented as a result. No doubt, but the maximum

the authors can say is that the results are consistent with what already known. In a way this should be taken for granted. What else? The same is true for the following paragraph.

127-132 – the comparison between the east side and west one of an ocean is not so simple. The whole interpretation is by far too simplistic.

139-140 – this kind of explanation would hardly be suitable for a first year course. There is no need of quoting Hwang or any other paper for this.

140-145 – all this discussion avoids the point if the wind is blowing offshore or onshore. If towards the coast, there no difference between swell and wind sea attenuation.

The authors recognize that the trivial explanation reported by the reviewer in the previous points is not needed. The two paragraphs concerning this and the previous Figure 4 are therefore removed. In the new section 2.2 a reference is put to previous studies to mention that we are consistent with what already known. Moreover, we underline the novelty coming from the observational knowledge presented in this study and compare the difference of offshore and coastal values concerning the seasonality with the differences seen in mean SWH in the previous section.

148-149 – there is nothing to confirm. Given that power depends on H^2T , the predominance of H is obvious

We agree with the reviewer. The sentence has been rephrased as follows: "...exhibiting the expected spatial variability consistent with the mean SWH..."

152-155 – I wonder about this statement. In a random sampling (in this case random with respect to the sea conditions) the mean should not depend on the sampling frequency. What varies are the confidence limits of the result. The matter would be different if looking for extremes.

We agree with the reviewer that the sentence was rather speculative. We have deleted it in the new version.

Figure 6 – a few things. Here we are zooming on an area to see the details. However, the vertical scale spans 10 degrees, 1111.11 km. If the details the reader should look at are at 30 km or less close to coast, this will correspond to 1/37 of the vertical scale of the figure, or less. This appears by far too little. In any case, if I interpret properly the figure, it looks to me that in the altimeter results (central part of the figure) the blue colour extends offshore more than the background that I interpret as what computed by Hemer et al.

We agree with the reviewer and in order to improve the quantification of the results and focus in the coastal zone, we produced a new subfigure showing bias and standard deviation between model and altimetry results. Moreover, we added the following discussion: "The agreement between this model and our derivation from the coastal altimetry data, shown in Figure 5a, is quantified in Figure 5b, where the mean and standard deviation of the differences between altimetry-derived and model-derived results is plotted with respect to the distance to coast, binned every 3-km. The mean bias is below 1 KW/m in the first 60 km from the coast, with a maximum standard deviation of about 5 KW/m close to the coast. Further away, altimetry tends to slightly underestimate the flux, but the mean bias is on average below 4 KW/m always below regardless of the distance to coast, i.e. less than 13% of the modelled wave energy flux."

Table 2 – these large scale values are of little help given the enormous variability within each indicated area. Also, as previously, there is no point in specifying the second decimal digit of a percent.

We agree with the reviewers that these are large scale values, but we still think that it is a useful way to summarise our findings. This is also in accordance to the method used by Reguero et al. (2015) to summarise findings about coastal wave energy flux and therefore we think there is a value in the comparison between the two studies to assess the progresses made. We agree with the reviewer concerning the second decimal digit of the %, nevertheless the third coloumn now reports the absolute differences, as suggested by another reviewer.

181-182 – this is not a general rule. It depends on the bottom topography, general geometry, et al. Last but not least, if the wind is blowing offshore or onshore.

As previously reported, we consider the cases and the causes of an increased mean SWH when approaching the land by discussing the new Figure 2

185-189 – these average conclusions are of little help or use given the spreading of the results, especially for energy flux.

In order to take into account the spreading of the results, we now plot also the distribution around the polynomial interpolation, i.e. the 95% confidence interval. The reviewer will notice that both in the case of the mean SWH and in the case of the energy flux the coastal attenuation w.r.t. the offshore point is verified within the 95% confidence interval. We added the following text “The global coastal attenuation is verified with a confidence level of 95% for both mean SWH and average energy flux. This is not true for the amplitude of the annual cycle, whose difference between coastal and offshore values has a wider spread.”

...-197 – see comment at the beginning. Datasets exist with 14 km resolution for the recent few years, slightly more coarse for the full ERA5 period.

The reviewer is right and the paragraph was rather referred to what is currently available in terms of evaluations of global estimations of wave power. In order to specify this, we added the following sentence: “Given the global observational representation of the coastal attenuation provided in this study, studies of global wave power shall be therefore updated using the latest models at higher resolution.”

217 – here we go back to my comment at the beginning- There are trivial explanations for wave oceanography, and professional ones for altimetry and related analysis. Given some trivial arguments about wave physics, for the benefit of the oceanographic readers (the ones more interested in the results), possibly a little explanation of the ALES algorithm could be useful.

A further paragraph is now added to further clarify the general procedure followed by ALES when fitting the waveforms:

“In the ALES retracker, the retracking of each waveform is performed in two passes. A first pass looks at the rising portion of the waveform (called leading edge) and provides a rough estimate of SWH from the slope of that portion. This estimate is then entered into an algorithm that selects the sub-waveform (i.e., sets the width of the fitting window over which a fitting is performed in the second pass). The dependence on the SWH is necessary to maintain the same level of precision achievable in the open ocean using a full-waveform retracker, given the direct relationship between sea state and noise of the retrieval.”

Figure 8 – horizontal units are missing

Corrected

256 – do I understand correctly that the authors derive one value every 350 m?

As previously stated: “The SGDR contain returned radar echoes, called waveforms, at a 20-Hz rate (corresponding to a distance of about 350m). Routinely in the standard product, a functional form (the Brown-Hayne model) is fitted to the waveform in a process called retracking.”.

And subsequently: “In order to decrease the noise of the high-frequency retrievals, standard altimetry data are routinely averaged at a 1-Hz rate (approximately one waveform every 7 km).”

Therefore, the reviewer understands correctly: one value every 350 m is estimated. In order to filter the noise, these retrieval filtered in blocks of 7 km.

261-262 – what else?

following – it appears complicated in the text, but it is fitting a sinusoid plus a trend. I agree that the iterative procedure is nice, but I wonder if it was really required. For instance, given the approximation of the data, which would be the differences with respect to taking a simple fit or even more simply the differences between the sequential max and min minimal values?

We provide here a clarification for the reviewer. The iterative procedure is necessary to avoid that the uncertainties in the estimation of the annual cycle are wrongly estimated using auto-correlated functions. We agree with the reviewer that there are different ways to account for autocorrelation. The Prais-Winsten method is one of them. To specify whether the iterative procedure is required, we show as an example the case of one time series (upper plot) which is auto-correlated at Lag-1 (central plot) and that undergoes the iterative procedure. The lower plot shows that after the application of Prais-Winsten the autocorrelation of Lag-1 (and subsequent lags, which were also previously not significantly autocorrelated) lowers down to a statistically not significant level. This is achieved in this case after 3 iterations. Of course we agree with the reviewers that not each and every time series at each 1-Hz point shows a significant auto-correlation. In that case, the numbers of iteration needed is indeed zero.

287 – “up to about 3 km” – I have a general question – going very close to coast implies probably more outliers, especially in a rough sea (a wider reflection zone). Through the described “cleaning” and selection procedure, this could affect the coastal values, hence statistics.

Independently of the paper I take this chance for a general question about which I ask an explanation by the authors. I have been using altimeter data for a long while of course, typical interval 7 km at 1 s interval. I understand each value is the average of 20 samples at 20 Hz uniformly distributed along the 7 km. Once these 20 data are averaged into a single value at a specified position, where is this position located along the 7 km? While I had this question for a while, I was wondering about it reading in this paper about going officially till 3 km off the coast. Thank you.

The procedure described in section 4.1.2 is exactly defined to avoid outliers and generate reliable 1-Hz averages of the data. For example, if we had to use 1-Hz point data closer to the coast, for example 1 Km, then there would be too many outliers that would prevent the computation of a reliable 1-Hz average. Moreover, the Jason missions "loses" some data in the transition from land to sea, due to an adjustment of the observation window, which means that closer than 3 km from the coast we would miss some 20-Hz retrievals, as the reviewer correctly notices. The comparison with the buoys in the study, including the figure added in the new version of the article, shows that the "cleaning and selection procedure" adopted is sufficient to observe the requested parameter (mean significant wave height) over the selected time scale.

We added the following paragraph to clarify the issue of the position of the 1-Hz point:

“It has to be noted that the 1-Hz SWH value along the track is nominally located at the centre of a segment of 20 20-Hz measurements and therefore is affected by the SWH retrievals located up to about 3.5 km before and after the nominal along-track point. In this study, a 1-Hz average is computed only if after the outlier procedure there are at least six valid 20-Hz measurements in the 1-Hz block. In this study, a 1-Hz average is computed only if after the outlier procedure there are at least six valid 20-Hz measurements in the 1-Hz block.”

289-290 – this is connected to a previous comment. The no doubt much lower number of altimeter data at buoy location will lead to less accurate results, but it should not affect the mean value in itself. This granted that the time of the day is not significant.

296 – at this level of sophistication 30 km can be relevant, especially close to coast where conditions are likely to change at shorter space interval. Indeed, the authors choose a max 15 km difference. Given the distance between adjacent satellite tracks (200 km or more), how general can the results be considered? This should be discussed.

We are aware of the fact that the scales of variation of the parameters that we are reporting are very small. At the same time, the comparison with the buoys, despite the fact that we are using the widest global dataset available, is limited, due to the small amount of in-situ data and the distance among the tracks. In order to expand the validation in this review, we have added two figures: a comparison of standard data vs our reprocessed dataset when approaching the coast in terms of correlation against the buoys (Figure 8) and a quantitative comparison in terms of wave energy flux against the high resolution model in Australia (Figure 5b). Concerning the generality of the results, we have added the level of confidence in Figure 6, in order to assess that the global average attenuation of SWH and energy flux is indeed statistically relevant. We believe that, given that in-situ data are on a fixed location and cannot compare offshore and coastal data of the same area, one of the interesting points of our research is that we bring observational evidence of processes that can be otherwise only modelled.

300 – “excellent agreement”. I never appreciate the qualitative appreciation, especially self-ones by authors. Results should be reported in a quantitative way, defining them useful or not for a specific purpose or another. In the present case, going back to what said above about Figure 6. I would have appreciated, and it would have been much more realistic, to zoom at the scale we are dealing with, 30 km or so, e.g. on the central part of Figure 6, and compare on this suitable zone the Hemer and altimeter results. From the figure I derive they are not so coincident.

We have eliminated the qualitative appreciation. To report results in a quantitative way, we added a new subfigure and the following discussion: “The agreement between this model and our derivation from the coastal altimetry data, shown in Figure 5a, is quantified in Figure 5b, where the mean and standard deviation of the differences between altimetry-derived and model-derived results is plotted with respect to the distance to coast, binned every 3-km. The mean bias is below 1 KW/m in the first 60 km from the coast, with a maximum standard deviation of about 5 KW/m close to the coast. Further away, altimetry tends to slightly underestimate the flux, but the mean bias is on average below 4 KW/m always below regardless of the distance to coast, i.e. less than 13% of the modelled wave energy flux.

Other qualitative statements throughout the manuscript have been removed and substituted where needed by further validation and quantification.

313-316 – is this true also in areas (most of the ocean coasts) with mixed wind sea - swell?

This is dependent on the spectral shape of the waves, and will vary. However these assumptions are used widely. Any representation of wave energy flux based on a single wave period value inherently makes some assumption on the spectral shape – this is no different, and as pointed out previously, is consistent with prior similar studies such as Young et al. 2013.

326 – “waves ..., generated locally”. The argument is not posed properly. Are we talking about offshore or inshore blowing wind? If towards the coast, waves are continuously generated till the coast. If from the coast, we go back to the point I mentioned above.

We are not certain of the point the reviewer is trying to make in this instance, but we believe that the reviewer’s views concur with our assumption: We are saying that we make an assumption that wave period remains constant through our calculation, but in reality this will be modulated due to local generation. Given that the global reanalysis that we use is distributed with a grid of half a degree, the local generation that can modulate the wave period at smaller scales cannot be considered (as opposite to the local changes of SWH, which are captured by the altimeter). The quantitative comparison against the Australian Wave Energy Atlas (added in this review in Figure 5b), which has a resolution of approximately 4km, should prove that this assumption does not hinder the estimation of the average wave energy flux, given that the mean bias found is below 1 KW/m in the first 60 km from the coast.

331 – same comment as above about Figure 6.

The statement is now rephrased as “A verification of this computation in absolute terms is provided regionally against model data in Figure 5b”. Note previous comment on the quantitative comparison now added in the subplot.

Conclusions

Basically the valuable part of the work done by the authors is the extension closer to coast of the results derivable from the Jason 1-2 Hs data. This is remarkable. There is some validation, but in my opinion not done in an objective way (e.g. Figure 6, if I have interpreted properly its significance).

We have now performed the objective (quantitative) validation related to Figure 6 (now Figure 5) and added additional validation with respect to standard altimetry data vs buoys in Figure 8

The so-called results obtained in the open ocean are obviously a repetition of what is presently common knowledge.

There is a lack of proper discussion and analysis of the relationship between wind and generated waves in the various possible situations. There is no discussion of the possibility of inshore and offshore blowing wind.

In the revised version we provide greater attention to the range of wave propagation and growth scenarios which can occur in the nearshore zone, presenting case study scenarios of (i) a scenario where wave heights are greater offshore and lower nearshore owing to the attenuation of swell propagating landwards towards the coast; (ii) a scenario with greater wave heights offshore than nearshore as a result of fetch-limited wave growth associated with offshore blowing winds; (iii) a scenario where wave heights nearshore are greater than wave heights offshore as a result of wave growth associated with onshore blowing winds. Further examples illustrate the effects of sheltering by islands. These case studies seek to illustrate a subset of the many scenarios that lead to differences

between the nearshore and offshore wave fields that be investigated further, enabled by the availability of our global dataset.

My final suggestion is to reject the paper encouraging a resubmission following the above suggestions and criticism. However, I strongly recommend that the authors look for the help of an oceanographer to deal in a more apt way with the physics of the marine part. If following this suggestion, for the benefit of the authors (but it may be already the approach by Nature Communications) I suggest to maintain the same editor and reviewers.

Luigi Cavaleri

Reviewers' comments:

Reviewer #1 (Remarks to the Author):

One of the most widely acknowledged limitations of using altimetry data for monitoring waves is their inaccuracy in coastal areas, due to alteration of the waveforms when land enters the radar footprint. The ALES algorithm applied to altimetry data, allows a characterisation of the wave attenuation in the coastal zone. This is a powerful tool for measuring wave characteristics and estimating wave energy fluxes near the coast, on a large scale. Therefore this presents a potentially wide interest for a range of applications, such as coastal erosion and protection, model calibration, wave energy site development, or coastal maritime traffic safety.

The article presents an application of the ALES algorithm to the characterisation of ocean waves attenuation in the coastal zone, from altimetry data (Jason-1 and Jason-2 missions). It gives a useful observation of the significant wave height and energy flux at a global scale both offshore and nearshore. Such a coverage can only be achieved by satellite altimetry. The application of the ALES retracker algorithm, detailed in previous papers from the same lead author, gives more reliable data close to the coast than previous altimetry data. The data is advantageously compared with wave buoy data. An estimate of the wave energy flux along the world coastal zones is presented, which gives an invaluable reference for wave data.

Reviewer #2 (Remarks to the Author):

Review of the NCOMMS manuscript "Global coastal attenuation of wind wave observed with radar altimetry" by Passaro et al.

I am afraid I am still rather sceptic regarding the present manuscript. Indeed the authors made an effort in improving the readability of the manuscript and correcting some of the major flaws. Nevertheless I still find the findings rather trivial, and a major mistake is still in place. For that matter I am rejecting the manuscript.

I make some comments below, and answer some of the authors' comments after that.

L26: reference missing after atmosphere.

L27: Ardhuin et al. (2019) is far from the best reference here. This is defined in Munk (1944):

Munk, W. H., 1944: Proposed uniform procedure for observing waves and interpreting instrument records. Scripps Institution of Oceanography, Wave Project Rep. 26, 22 pp.

L30: "Such climatology"? Which climatology? The wave energy resources are not only a function of wave heights, hence this sentence is misleading. You cannot build a climatology with y years of observations, with such a low temporal resolution.

L61 and L65. Why are the Jason-1 and Jason-2 missions chosen in the present study, and why are they "unprecedented". Have you looked at the most recent altimetry observations of Sentinel 3?

L77: mean? Annual mean? (This comments applies across the manuscript.)

L81: well understood climatology? I am afraid what you present on Figure 1a cannot be seen as a climatology. Besides that, what is meant with "well understood"?

L82: highest SHWs should be the highest mean SWHs (this applies to several parts of the manuscript).

L93: coastal attenuation of what? SWH?

L109: wave energy is different from wave height. Where do you see refraction in Figure 2b?

L130-132: sentence starting with "Open ocean..." comes along with my assessment of this manuscript, as I have mentioned in my previous review, as well as here. I hope the authors can understand that "resemble" is not exactly a very accurate statement, besides, why wouldn't it "resemble". If you take one year of SWH altimetry observations I am sure it will also resemble. I am afraid I do not understand what the authors want to convey with this sentence.

L143.144: What makes you conclude (suggest) that there is an independence between the offshore and the coastal wave climates? (1) You are far from having enough elements to conclude this, and (2) from the (few) you present it seems exactly the opposite.

L146: not "your" data set (this applies across the manuscript), but the "data set used in the present study". Besides that you are using (wrongly in my opinion) satellite altimetry and reanalysis data, hence being more specific wouldn't hurt at this level.

L147: again, why should they agree? How much do they agree?

L169: again our not correct, and estimates might not be very correct either.

L174: between the model or the model results? And what model results are we talking about? A regional hindcast? What is coastal altimetry?

L178: why underestimate? What makes you conclude that the model results are better? You are comparing outputs, not evaluating skills here.

L180: if this one is modelled, what is the one you compute using observations and reanalysis?

L184: mean state of what? Seasonality of what?

L185-186: don't understand the statement starting with "that could...", besides that the simulations in Perez et al. (2017) cannot be considered as high resolution.

L196: what are the differences between model and reanalyses (and not reanalysis) in this context?

L338: dependency on wave period and wave height.

From here onwards I provide some comments to the authors' rebuttal #1.

Q1 Why in the global maps you chose not to grid the altimetry along tracks and instead plot the contours in the along track data (as in figure 1a, for example)? Never seen it. Not correct. You have to grid the altimetry data, averaging it into boxes (1 or 2 dgrs) and then produce the contours. You also fail to explain your averaging technique properly.

It is not clarified in the review which would be the fundamental error in the core of this study according to the reviewer. From the sentence above, we assume the reviewer thinks that the fact that altimetry time series are created along the track instead of gridding is a fundamental error. We do not think that this opinion is justified. Surely we agree that in the standard literature, as reported in the Introduction of this paper, the data are averaged into boxes (1 or 2 dgrs). The reviewer will agree with us that for the specific purpose of this paper, which is to analyse differences within 30 km of the coast, this methodology would not make sense, since a 1-degree grid has a spacing of several tens of kms.

Q2 For the purpose of the coastal study I agree, that gridding the data might not be the best approach. Nevertheless, for the global annual mean SWH, considering the time resolution of the satellite altimetry data, that is not the best approach. Would you say that the global mean SWH patten is as you present it in Figure 1a? Therefore, what is the goal of Figure 1a?

(The fundamental error was not here, as you latter saw in the review.)

Q1 Also the ERA5 wave reanalysis spatial resolution is not 0.5 dgr.

We attach here the table available from the link that describes ERA5 wave reanalysis data, from which data are downloaded, as reported in the Acknowledgements. According to this table, if we are wrong, then also the official description is wrong. Nevertheless we acknowledge that the original resolution of the model used for the reanalysis may be higher [1], therefore we correct the sentence as follows: "...provided by the ERA5 reanalysis on a 0.5 x0.5 grid..."

We also report the text of a personal communication from Jean Bidlot, coauthor of the ERA5: "...part of the confusion might come from the fact that ERA5 is NOT using the same resolution as our operational forecast system...Our operational forecast wave products are produced on a native grid with a resolution ~14km, which are made available to users on 0.1x0.1 or 0.125x0.125 degree gridswhereas, ERA4 is produced on a native grid with a resolution ~40km, made available via the CDS on 0.5x0.5 degree grid" [1] <https://www.ecmwf.int/en/about/media-centre/news/2019/upgrade-boost-quality-ocean-wave-forecasts>

Q2 It seems, as I said, that you confirm your lack of knowledge on some basic matters. The ERA5 reanalysis is produced with a 31 km (atmosphere) and 0.36 dgrs (waves) (see Herbach et al. 2020). When reanalysis data is downloaded from the ECMWF site, you have a choice to interpolate the data to a certain degree, especially the atmospheric part that is produced using a global spectral model. Hence... The resolution of the waves output of ERA5 is 0.36 dgrs, and not 0.5 dgrs.

Q1 You cannot compute the climatological mean wave energy flux using altimetry SWH and ERA5 reanalysis mean wave periods. This is fundamentally wrong and cannot be done.

The wave energy flux is now computed for every point of the altimetry track using the instantaneous estimation of SWH and the closest value in time of the relevant wave period from the ERA5 reanalysis (which is distributed with a 3-hour time resolution). A similar approach (i.e. coupling wave periods from the model Wavewatch III with SWH from altimetry) was used in previous studies such as Young et al., 2013. Finally, the validation of our approach against a fully modelled approach as in Hemer et al. is now provided in Figure 5b.

Q2 I am afraid your study maintains a flaw that leads me to reject it. You cannot produce the climatological mean wave energy flux using observations and reanalysis. Climatological statistics are highly sensitive, and by doing this you can easily be getting wrong results. Additionally both the altimetry measurements and the global reanalysis are not the most perfect products at the coast in terms of reliability. Simple: why don't you use reanalysis only? Try this simple exercise: compute the wave energy flux using just ERA5 at the coast, and you might get some surprises compared to your results. I disagree that Young and Babanin (2010) used the same approach. The time scales are different, and they use it for deep waters, on a case study basis.

Reviewer #3 (Remarks to the Author):

See attached file.

The paper has improved a lot, but still some remarks about the errors associated to the chosen deep water approach for the mean wave energy flux should be indicated. As SWH satellite information near the coast is a valuable instrumental information for calibration and validation of downscaled reanalysis propagation models, the validation methodology of the satellite data with wave buoy data should be improved.

Reviewer #4 (Remarks to the Author):

Comments on the paper

"Global coastal attenuation of wind-waves observed with radar altimetry"

by Marcello Passaro, Mark A.Hemer, Christian Schwatke, Denise Dettmering, and Florian Seitz

The following are my comments on the second version of this very interesting paper. I am pleased to acknowledge that all the points I had raised in my first review have been properly addressed. There is no doubt that including a first class oceanographer as Mark Hemer has provided the necessary oceanographic expertise that was obviously lacking in the first submission. This has not only corrected the unprecise statements and conclusions of the first version, but it has also provided new brilliant insights that give increased value to the overall product. I strongly suggest to accept the paper for publication, but only after some simple, but, some at least, necessary corrections that I list below. I do not need to see again the fully corrected version of the paper.

-(line) 102 – I suggest "... altimetry (colour scale) with vectors.."

- 103 – I suggest to mention the location (Alaska) of panel 2a also in the text. Presently it is only in the caption.

-Figure 2 (caption) – I suggest ".. along the altimetry tracks (colour scale) in Alaska"

- 134 – "... amplitude are shown ."

- 128 and onwards – In my view there is a repeated error in referring to the differences between offshore and coastal values. I cite various points, but I am not sure I have picked up all of them. In Table 2 the differences are 'offshore – coastal'. Citing the various lines I spotted: 129) "relative to the offshore estimate". Usually this means 'coastal –offshore'. 150-151 - same comment. Caption Figure 3: although here you talk about absolute values, the original difference is between offshore and coastal values.

- Figure 3, caption – "marked with a black cross. The point ..". Of course this is a left-over of the previous version of the Figure. It needs to be updated to the new version of the Figure.

- 228 and following – If I understood correctly, I have a question: with 'pass', do you mean 'step'? If so, I suggest 'step'. 'Pass' has too much to do with altimeters.

- 344-345, formula (5) – $SWH \diamond SWH$

Very good work and I congratulate the authors for the very nice piece of work. As I have already specified, no need to see the text again after the above trivial corrections.

Luigi Cavaleri

Reviewer #3 additional file:

Wave Satellite Passaro. 2nd rev.

Figure 3. There are not black crosses in the figure.

Pages 18-19. Validation of satellite SWH against buoys SWH should be carried out very carefully for these buoys located in intermediate waters. In intermediate water depths, the spatial gradient of SWH could be important and the comparison of buoy measurements with satellite track points located several Km away could give important differences just because differences in wave propagation.

A much more precise methodology would be first to propagate the corresponding global reanalysis sea state directional spectrum from offshore to the buoy and satellite track point locations (taking into account local winds and bathymetry), and use the buoy-satellite SWH propagation coefficient to transpose the buoy measurement to the satellite track point (or vice versa).

Page 20. Wave energy flux equation (5).

As indicated in the text, the mean energy flux is calculated assuming the deep water approach (5) could be applicable to intermediate waters (up to 3 km to the coast).

The wave energy flux, of a monochromatic wave train (H, T) is given by:

$$W = \frac{1}{8} \rho g H^2 C_g \quad (1)$$

Where C_g is the wave group celerity $C_g = \frac{c}{2} \left(1 + \frac{2kh}{\sinh(2kh)} \right)$

And c is the wave celerity: $c = \frac{L}{T}$, $k = \frac{2\pi}{L}$ is the wave number and L is the wave length, that is obtained solving the dispersion relationship:

$$L = L_0 \tanh \left(\frac{2\pi h}{L} \right)$$

Where $L_0 = \frac{gT^2}{2\pi}$

In deep waters $C_{g0} = \frac{c_0}{2}$ and $c_0 = \frac{gT}{2\pi}$, so equation (1) simplify to:

$$W_0 = \frac{\rho g^2}{32\pi} H^2 T$$

If the relative difference:

$$\frac{W - W_0}{W} * 100$$

Is computed in terms of the relative water depth, h/L , the following curve is obtained:

As can be seen, the deep water formulation underestimate the energy flux until $h/L \cong 0.1$, with a maximum underestimation of 16.64% for $h/L = 0.1909$. For relative water depths lower than 0.1, the deep water approximation overestimates the wave energy flux, and when the shallow water region is reached ($h/L < 0.05$), this overestimation is above 70%.

Concentrating on the intermediate water depths above $h/L > 0.1$ the water depth at which the maximum underestimation is obtained can be calculated for each wave period, obtaining the following curve:

As can be seen, for typical ocean wave periods (6 – 20 s), the water depths for which the deep water approximation produces the maximum underestimation of the wave energy flux varies between 8.9 m for $T = 6$ s to 99.4 m for $T = 20$ s. Assuming that an average continental shelf

slope is around 1/400, the corresponding distances to the coast are between 3.6 and 39.8 Km well into the range of distances where the satellite SWH are obtained.

As the exact wave energy flux of each sea state:

$$W_{ss} = \rho g \int_0^{\infty} \int_0^{2\pi} C_g(f) \cdot S(f, \theta) d\theta df$$

cannot be obtained from the satellite information, at least a clear explanation about the error of using the deep water approach for computing the mean wave energy flux should be addressed in the paper.

It should be stressed that the SWH satellite information near the coast is a very useful instrumental information for calibration and validation of downscaled wave reanalysis. This gives maximum importance to the aforementioned validation of the satellite SWH using wave buoys measurements.

Once the validated nearshore satellite information is used to calibrate and validate the downscaled reanalysis, these reanalysis can be used for any engineering application on the coast (among them the wave energy flux distribution).

Reviewers' comments:

Reviewer #1 (Remarks to the Author):

One of the most widely acknowledged limitations of using altimetry data for monitoring waves is their inaccuracy in coastal areas, due to alteration of the waveforms when land enters the radar footprint.

The ALES algorithm applied to altimetry data, allows a characterisation of the wave attenuation in the coastal zone. This is a powerful tool for measuring wave characteristics and estimating wave energy fluxes near the coast, on a large scale. Therefore this presents a potentially wide interest for a range of applications, such as coastal erosion and protection, model calibration, wave energy site development, or coastal maritime traffic safety.

The article presents an application of the ALES algorithm to the characterisation of ocean waves attenuation in the coastal zone, from altimetry data (Jason-1 and Jason-2 missions). It gives a useful observation of the significant wave height and energy flux at a global scale both offshore and nearshore. Such a coverage can only be achieved by satellite altimetry. The application of the ALES retracker algorithm, detailed in previous papers from the same lead author, gives more reliable data close to the coast than previous altimetry data. The data is advantageously compared with wave buoy data. An estimate of the wave energy flux along the world coastal zones is presented, which gives an invaluable reference for wave data.

We warmly thank the reviewer for the support.

Reviewer #2 (Remarks to the Author):

Review of the NCOMMS manuscript "Global coastal attenuation of wind wave observed with radar altimetry" by Passaro et al.

I am afraid I am still rather sceptic regarding the present manuscript. Indeed the authors made an effort in improving the readability of the manuscript and correcting some of the major flaws.

Nevertheless I still find the findings rather trivial, and a major mistake is still in place. Fro that matter I am rejecting the manuscript.

I make some comments below, and answer some of the authors' comments after that.

L26: reference missing after atmosphere.

We have added the following reference:

Cavaleri, L., Fox-Kemper, B. and Hemer, M., 2012. Wind waves in the coupled climate system. *Bulletin of the American Meteorological Society*, 93(11), pp.1651-1661.

L27: Ardhuin et al. (2019) is far from the best reference here. This is defined in Munk (1944):

Munk, W. H., 1944: Proposed uniform procedure for observing waves and interpreting instrument records. Scripps Institution of Oceanography, Wave Project Rep. 26, 22 pp.

We have referenced a review paper in which the reader can also find the connection between the physical variable defined and the measurement technique used in this study. It is clear that it is not the first paper defining that. Moreover, we think the journal prefers references to peer-reviewed papers.

L30: “Such climatology”? Which climatology? The wave energy resources are not only a function of wave heights, hence this sentence is misleading. You cannot build a climatology with y years of observations, with such a low temporal resolution.

This sentence was already present in the previous review of the article, was accepted by all other reviewers and the point was not raised by this reviewer in the previous submission. Nowhere have we said that the wave energy resources are ONLY a function of wave heights, so we do not think the sentence is misleading.

L61 and L65. Why are the Jason-1 and Jason-2 missions chosen in the present study, and why are they “unprecedented”. Have you looked at the most recent altimetry observations of Sentinel 3?

The observations from Sentinel-3 span only 4 years yet, with a repetition cycle of 27 days. This is clearly less adapt for an along-track analysis than the 15-year-long record with 10-day repetitions, which is the subject of this study. The quality and quantity of the coastal data reprocessed using Jason is indeed unprecedented, as we demonstrate with the validation. .

L77: mean? Annual mean? (This comments applies across the manuscript.)

“Mean SWH” is correct. It is the mean measured over the time period studied. Moreover, the use of this term was already present in the previous review of the article, was accepted by all other reviewers and the point was not raised by this reviewer in the previous submission.

L81: well understood climatology? I am afraid what you present on Figure 1a cannot be seen as a climatology. Besides that, what is meant with “well understood”?

The reviewer himself argued in the first review that the paper should have been freed from trivial, known statements, and that there is nothing new in demonstrating that this dataset says the same thing of already existing dataset for already known scales (such as the global large scale patterns).

Figure 1a helps the reader to observe that the use of along-track data in computing the global mean SWH, which is necessary here in order to observe the mean SWH in the coastal zone, is also a valid approach to observe the typical, known, mean global SWH patterns (referenced through Young, 1999, therefore well understood).

L82: highest SHWs should be the highest mean SWHs (this applies to several parts of the manuscript).

Although it is already clear from the title of the section that we are talking about mean SWH when referring to the results presented and although we would have liked to avoid the repetitions, we have now added “mean” to “SWH” in this location and in three others of this section.

L93: coastal attenuation of what? SWH?

This is clear from the previous line “Table 1 shows the regional average attenuation of SWH”.

L109: wave energy is different from wave height. Where do you see refraction in Figure 2b?

We have adjusted this text to clarify the scenario presented in Figure 2b, where dissipation of wave energy is seen in the altimeter derived wave height observations, and refraction of order 5 degrees is observed between the off-shelf and on-shelf locations in the reanalysis wave direction record.

L130-132: sentence starting with “Open ocean...” comes along with my assessment of this manuscript, as I have mentioned in my previous review, as well as here. I hope the authors can understand that “resemble” is not exactly a very accurate statement, besides, why wouldn't it “resemble”. If you take one year of SWH altimetry observations I am sure it will also resemble. I am afraid I do not understand what the authors want to convey with this sentence.

This sentence is needed to observe that the use of along-track data in computing the annual cycle of the SWH, which is necessary here in order to observe the behavior of the coastal zone record, is also a valid approach to observe the typical, known, open ocean pattern. Moreover, this sentence was already present in the previous review of the article, was accepted by all other reviewers and the point was not raised by this reviewer in the previous submission.

L143.144: What makes you conclude (suggest) that there is an independence between the offshore and the coastal wave climates? (1) You are far from having enough elements to conclude this, and (2) from the (few) you present it seems exactly the opposite.

We do not understand the comment of the reviewer. The reviewer himself notices that we have not “concluded”, but “suggested” this link. Moreover, if the whole paragraph is read, it is in our opinion evident that the general conclusion is what the reviewer refers as “the opposite”, i.e. “This attenuation of seasonality is largely consistent with a proportional attenuation of the coastal wave heights presented”.

The sentence extracted by the reviewer only refers to the locations where the opposite is observed, which is the smallest group, as the reviewer also notices (“Of the locations not displaying significant difference between the offshore and coastal seasonality, 33%”)

L146: not “your” data set (this applies across the manuscript), but the “data set used in the present study”. Besides that you are using (wrongly in my opinion) satellite altimetry and reanalysis data, hence being more specific wouldn't hurt at this level.

We have already explained why we do not consider that satellite altimetry and reanalysis data are wrongly used. We have also validated it. Finally, “our dataset” is a terminology that is fully justified in this case and commonly used in papers of the same journal family (just one example: <https://www.nature.com/articles/s41561-020-0618-x>, “Our dataset includes 10,276 individual valleys...”). Moreover, the first author of this paper is also the first author of the algorithm generating this dataset.

L147: again, why should they agree? How much do they agree?

The reviewer himself argued in the first review that the paper should have been freed from trivial, known statements, and that there is nothing new in demonstrating that this dataset says the same thing of already existing datasets for already known scales (such as the global large scale patterns). The global average energy flux is known and the sources where the reviewer can find information about that are reported in the same paragraph. Just as in the case of the global mean SWH field. The “how much” of interest in this paper are the coastal scales. A validation is performed in an area where high resolution data are available at the coast, such as the southern Australian coast (see figure 5).

L169: again our not correct, and estimates might not be very correct either.

We do not understand the statement “estimates might not be very correct either”. On which basis, given the validation offered in this paper? Moreover, this sentence was already present in the previous review of the article, was accepted by all other reviewers and the point was not raised by this reviewer in the previous submission.

L174: between the model or the model results? And what model results are we talking about? A regional hindcast? What is coastal altimetry?

The source has already been cited, nevertheless to be more specific on what we “are talking about” we added “[The Australian Wave Energy Atlas (Hemer et al., 2017; Hemer and Griffin, 2010)) presents the wave climate around the Australian continent at a resolution of approximately 4 km] and is based on a global implementation of the WAVEWATCH III (v4.08) hindcast, with a series of nested high-resolution computational grids in the Australian and South Pacific region.”

We also add “model results” instead of “model”.

L178: why underestimate? What makes you conclude that the model results are better? You are comparing outputs, not evaluating skills here.

Since we are talking about a comparison between two solutions, it is clear that by underestimation we mean that the estimates of the altimeter-derived solution are in that case lower than the reference we are using for the comparison.

L180: if this one is modelled, what is the one you compute using observations and reanalysis?

This same reviewer correctly requested in the first round of review not to mix the concepts of reanalysis and models. The solution of the Australian Wave Energy Atlas is based on a hindcast. A hindcast per definition refers to realization of numerical models in the past in which no data have been assimilated. Therefore it is correct to talk about it as the “modelled” solution. Since our solution is based on the use of observations and reanalysis, it cannot be defined as a modelled solution.

L184: mean state of what? Seasonality of what?

The word “SWH” is reported in the same sentence, therefore it is obvious that we are referring to it. Moreover, this sentence was already present in the previous review of the article, was accepted by all other reviewers and the point was not raised by this reviewer in the previous submission.

L185-186: don't understand the statement starting with "that could...", besides that the simulations in Perez et al. (2017) cannot be considered as high resolution.

"that could so far..." means that the cited parameters were not observed at this coastal scale from satellite altimetry before this study. The addition was made to answer a point of another reviewer (who wrote "In the paper of Pérez et al. (2017) The Global Ocean Waves Two database (GOW2) was presented. GOW2 is a wave hindcast based on WaveWatch III (WWIII) version 4.18 that divides the World Ocean in four regular meshes: the global mesh (0.5o by 0.5o), two regional meshes that cover the Arctic and Antarctic areas (0.25o latitude by 0.5o longitude) and the coastal mesh (0.25o by 0.25o). This coastal mesh includes all the points with water depth below 200 m and a surrounding area of 1.5o. These data bases have been calibrated and validated using all available instrumental data (satellite and wave buoys).")

The other reviewer has accepted the applied change to the text.

L196: what are the differences between model and reanalyses (and not reanalysis) in this context?

The term "reanalysis" is correct and used by ECMWF itself:

<https://www.ecmwf.int/en/research/climate-reanalysis>

Moreover, this same reviewer has suggested to use the term reanalysis...(to be continued)

L338: dependency on wave period and wave height.

This is clear from the formula and the objective was to report how we dealt with the wave period. Nevertheless we welcome the reviewer's comment and changed the sentence accordingly.

From here onwards I provide some comments to the authors' rebuttal #1.

Q1 Why in the global maps you chose not to grid the altimetry along tracks and instead plot the contours in the along track data (as in figure 1a, for example)? Never seen it. Not correct. You have to grid the altimetry data, averaging it into boxes (1 or 2 dgrs) and then produce the contours. You also fail to explain your averaging technique properly.

It is not clarified in the review which would be the fundamental error in the core of this study according to the reviewer. From the sentence above, we assume the reviewer thinks that the fact that altimetry time series are created along the track instead of gridding is a fundamental error. We do not think that this opinion is justified. Surely we agree that in the standard literature, as reported in the Introduction of this paper, the data are averaged into boxes (1 or 2 dgrs). The reviewer will agree with us that for the specific purpose of this paper, which is to analyse differences within 30 km of the coast, this methodology would not make sense, since a 1-degree grid has a spacing of several tens of kms.

Q2 For the purpose of the coastal study I agree, that gridding the data might not be the best approach. Nevertheless, for the global annual mean SWH, considering the time resolution of the satellite altimetry data, that is not the best approach. Would you say that the global mean SWH patten is as you present it in Figure 1a? Therefore, what is the goal of Figure 1a?

(The fundamental error was not here, as you latter saw in the review.)

Figure 1a helps the reader to observe that the use of along-track data in computing the global mean SWH, which is necessary here in order to observe the mean SWH in the coastal zone, is also a valid approach to observe the typical, known, mean global SWH patterns (referenced through Young, 1999).

Q1 Also the ERA5 wave reanalysis spatial resolution is not 0.5 dgr.

We attach here the table available from the link that describes ERA5 wave reanalysis data, from which data are downloaded, as reported in the Acknowledgements. According to this table, if we are wrong, then also the official description is wrong. Nevertheless we acknowledge that the original resolution of the model used for the reanalysis may be higher [1], therefore we correct the sentence as follows: "...provided by the ERA5 reanalysis on a 0.5 x0.5 grid..."

We also report the text of a personal communication from Jean Bidlot, coauthor of the ERA5: "...part of the confusion might come from the fact that ERA5 is NOT using the same resolution as our operational forecast system...Our operational forecast wave products are produced on a native grid with a resolution ~14km, which are made available to users on 0.1x0.1 or 0.125x0.125 degree gridswhereas, ERA4 is produced on a native grid with a resolution ~40km, made available via the CDS on 0.5x0.5 degree grid"

[1] <https://www.ecmwf.int/en/about/media-centre/news/2019/upgrade-boost-quality-ocean-wave-forecasts>

Q2 It seems, as I said, that you confirm your lack of knowledge on some basic matters. The ERA5 reanalysis is produced with t a 31 km (atmosphere) and 0.36 dgrs (waves) (see Herbach et al. 2020). When reanalyses data is downloaded from the ECMWF site, you have a choice to interpolate the date to a certain degree, especially the atmospheric part that is produced using a global spectral model. Hence... The resolution of the waves output of ERA5 is 0.36 dgrs, and not 0.5 dgrs.

We do not understand the reviewers' point. We already corrected this point by saying that 0.5 degrees is not the resolution of the reanalysis, but the grid size with which the data are provided.

Q1 You cannot compute the climatological mean wave energy flux using altimetry SWH and ERA5 reanalysis mean wave periods. This is fundamentally wrong and cannot be done.

The wave energy flux is now computed for every point of the altimetry track using the instantaneous estimation of SWH and the closest value in time of the relevant wave period from the ERA5 reanalysis (which is distributed with a 3-hour time resolution). A similar approach (i.e. coupling wave periods from the model Wavewatch III with SWH from altimetry) was used in previous studies such as Young et al., 2013. Finally, the validation of our approach against a fully modelled approach as in Hemer et al. is now provided in Figure 5b.

Q2 I am afraid you study maintains a flaw that leads me to reject it. You cannot produce the climatological mean wave energy flux using observations and reanalysis. Climatological statistics are highly sensitive, and by doing this you can easily be getting wrong results. Additionally both the altimetry measurements and the global reanalysis are not the most perfect products at the coast in

terms of reliability. Simple: why don't you use reanalysis only? Try this simple exercise: compute the wave energy flux using just ERA5 at the coast, and you might get some surprises compared to your results. I disagree that Young and Babanin (2010) used the same approach. The time scales are different, and they use it for deep waters, on a case study basis.

ERA5 cannot be used at the coast as we are doing with along-track altimetry, since the resolution of ERA5 is (even following the indication of the reviewer) 0.36 degrees, i.e. more than 30 km, while the differences we want to highlight are below this spatial scale. The comment of the reviewer is misleading, first he says that reanalysis is not the perfect product at the coast; secondly he proposes to use reanalysis only. We have already quantified the error we obtain in the mean values by using the deep water approximation. The reviewer seems to ignore the validation of our coastal product that we have reported in the paper, and defines a-priori that altimetry in general is not to be used at the coast, regardless of the years of progress in this field. Besides validation against in-situ data, which we have performed, we do not know what else would convince the reviewer. In this new version we have added a new ad-hoc validation subsection and adopted the approach of Nencioli & Quartly (2019) in using a high-resolution (1.5km) wave model reanalysis to assess whether the SWH time series at the buoy and altimeter locations vary coherently ($r^2 \geq 0.85$ and $\text{RMSD} \leq 0.3\text{m}$). Restricting our altimeter validation to those points shows yet better results for ALES.

The simple exercise proposed by the reviewer to test the climatology using reanalysis wave heights and periods to compute the energy flux is not a valid exercise – there is no doubt that the values will differ, as the sensitivity of biases in wave heights will be amplified by the square dependency of wave height in the flux calculation. This does not infer the observational derived value is incorrect, or the fully reanalysis derived value is incorrect, simply that there is a bias.

Reviewer #3 (Remarks to the Author):

See attached file.

The paper has improved a lot, but still some remarks about the errors associated to the chosen deep water approach for the mean wave energy flux should be indicated.

Firstly we would like to thank the reviewer for the positive approach and constructive comments.

We accept the remark of the reviewer and therefore we have added the estimation of the error associated with the deep water approach directly in the paper, adding a new section 4.5 "Errors due to deep water assumption". The reviewer will notice that, compared to the equation (1) reported in the attachment provided, we use the following expression for the wave energy density per unit area:

$$E = 1/16 * \rho * g * \text{SWH}^2$$

The difference lies in the fact that the 1/8 coefficient (adopted in the equation reported by the reviewer) is used in linear theory, using a monochromatic wave height, as the reviewer correctly reports. The 1/16 coefficient is used assuming linear theory, but for a complete spectrum, and uses a significant wave height, which is therefore more appropriate for our approach.

As SWH satellite information near the coast is a valuable instrumental information for calibration and validation of downscaled reanalysis propagation models, the validation methodology of the satellite data with wave buoy data should be improved.

We agree that altimeter validation in the coastal zone is more complicated because of the potential for significant spatial gradients in SWH. We have therefore added a new ad-hoc validation subsection and adopted the approach of Nencioli & Quartly (2019) in using a high-resolution (1.5km) wave model reanalysis to assess whether the SWH time series at the buoy and altimeter locations vary coherently ($r^2 \geq 0.85$ and $\text{RMSD} \leq 0.3\text{m}$). Restricting our altimeter validation to those points shows yet better results for ALES but has minimal impact on the assessment of the GDR values. We believe that this concept of "coherent areas" is clearer than propagating a wave spectrum adding in extra contributions for local winds and modelled bathymetry.

Reviewer #4 (Remarks to the Author):

Comments on the paper

“Global coastal attenuation of wind-waves observed with radar altimetry”

by Marcello Passaro, Mark A.Hemer, Christian Schwatke, Denise Dettmering, and Florian Seitz

The following are my comments on the second version of this very interesting paper. I am pleased to acknowledge that all the points I had raised in my first review have been properly addressed. There is no doubt that including a first class oceanographer as Mark Hemer has provided the necessary oceanographic expertise that was obviously lacking in the first submission. This has not only corrected the unprecise statements and conclusions of the first version, but it has also provided new brilliant insights that give increased value to the overall product. I strongly suggest to accept the paper for publication, but only after some simple, but, some at least, necessary corrections that I list below. I do not need to see again the fully corrected version of the paper.

We thank the reviewer for the support and the feedback

-(line) 102 – I suggest “... altimetry (colour scale) with vectors..”

Corrected as suggested

- 103 – I suggest to mention the location (Alaska) of panel 2a also in the text. Presently it is only in the caption.

Added “...showing a section of the Alaska's coast...” in the text

-Figure 2 (caption) – I suggest “.. along the altimetry tracks (colour scale) in Alaska”

Corrected as suggested

- 134 – “... amplitude are shown .”

Corrected as suggested

- 128 and onwards – In my view there is a repeated error in referring to the differences between

offshore and coastal values. I cite various points, but I am not sure I have picked up all of them. In Table 2 the differences are 'offshore – coastal'. Citing the various lines I spotted: 129) “relative to the offshore estimate”. Usually this means 'coastal –offshore'. 150-151 - same comment. Caption Figure 3: although here you talk about absolute values, the original difference is between offshore and coastal values.

In order to be fully clear on the order of the differentiation also in the text, we have corrected as suggested the following points (no other relevant cases were found), the line numbering refers to the previous submission in order to match the line numbering of the reviewer:

129: “the difference between coastal and offshore estimations (offshore-coast)”

150: “the difference between coastal and offshore measurements (offshore-coast)”

caption Figure 3, “difference between the coastal and the offshore amplitude”

- Figure 3, caption – “marked with a black cross. The point ..”. Of course this is a left-over of the previous version of the Figure. It needs to be updated to the new version of the Figure.

Thanks for spotting this typo. We changed with “black contour”

- 228 and following – If I understood correctly, I have a question: with 'pass', do you mean 'step'? If so, I suggest 'step'. 'Pass' has too much to do with altimeters.

We accepted the suggestion and changed the text accordingly

- 344-345, formula (5) – $SW H \diamond SWH$

The formula is correctly written in Latex, this is just an effect of the print

Very good work and I congratulate the authors for the very nice piece of work. As I have already specified, no need to see the text again after the above trivial corrections.

Luigi Cavaleri

REVIEWER COMMENTS

Reviewer #1 (Remarks to the Author):

The third draft of the manuscript entitled "Global coastal attenuation of wind-waves observed with radar altimetry" by Passaro et al. Is a clear improvement compared to the initial version, in terms of readability and content.

Just a quick note on some of the rebuttal comments from the authors: although I acknowledge that we have to draw a line at some stage, the fact that some comments on the second draft were not issued during the first review (on the initial draft) does not mean that they should be discarded, as reviewers can also miss some important questions the first time. For a journal such as Nature Communications, articles must be of high quality and therefore all justified comments have to be addressed. The only exception is if comments appear to be contradictory between the two revisions, in which case they are open to debate for clarification.

I understand the comments of reviewer #2 on lines 143-144, as the paragraph starting in line 139 with "Statistically significant differences [...]" does not read well: the explanation is a bit muddled, especially with the frequent occurrence of the terms significant and non-significant. An illustration of this is the sentence: "Of the locations not displaying significant difference [...] in the coastal zone". I strongly suggests that the authors re-read this paragraph and try to make it clearer.

The conclusion of this paragraph lies in the last sentence where it is deduced that the independence of offshore and coastal wave climate corresponds to fetch limited areas. I agree with reviewer #2 that the result sounds too qualitative and uncertain (the words 'mostly' and 'suggesting' are too vague and not conclusive enough). If this is an important result, then it should be better quantified and backed-up by a more rigorous argumentation, for instance by estimating the fetch at each site and making a statistical analysis with the significant difference. I realise that this could involve a substantial additional work, so I leave it to the authors to decide whether this is feasible (and the best method for it), and/or worth it.

Going to main contentious argument, I agree with reviewer #2 that you can get significant errors when calculating the energy flux with mix data source (SWH from altimetry data and wave period from ERA5), and I apology that I overlooked that in my first review. There is a good effort in section 2.3 in justifying the fit between the wave energy fluxes calculated from WAVEWATCH III hindcast and altimetry data. I was just puzzled by the magnitude of the standard deviations in Figure 5b, and wondered (i) if the mean difference was meaningful in this case, especially near the coast where the energy flux is lowest, and (ii) where these high standard deviations come from.

A comparison test with energy fluxes calculated purely with ERA5 as suggested by reviewer #2 would have been interesting in theory, but as the authors rightly pointed out, this may not easily be done considering the difference in grid resolution. Another avenue that could be investigated is by estimating the wave period from the backscattering coefficient and significant wave height, following the method proposed by Gommenginger et al. [Gommenginger C.P., Srokosz M.A., Challenor P.G., Cotton P.D. (2003) Measuring ocean wave period with satellite altimeters: a simple empirical model, *Geophys.Res.Letters*, 30, issue 22, DOI: 10.1029/2003GL017743], and successfully validated by Goddijn-Murphy et al. (2015) using Altika altimetry data against wave buoy records. The method was derived in deep water, so I am unsure how applicable this method can be near the coast, but as I said it may be worth investigating it and see how this compares with the values you used.

In conclusion, I still stand with my previous review where I considered that the subject and the amount of work put in this study justifies being published, but it is also important that the scientific methods and discussions are very rigorous for the high standards of a scientific journal such as Nature Communications.

There have been some significant improvements from the first draft including a more detailed comparison between model and data calculated energy fluxes, a validation against coastal wave buoys and a discussion on the deep water assumption. The use of the ERA5 period in the energy flux calculations remains questionable but there is no obvious better alternative and the authors made an extensive effort to mitigate

it by analysing the model hindcast versus the altimetry data.

After revising the readability of the paragraph starting line 139, as discussed earlier, I would be happy for the manuscript to be published.

Reviewer #3 (Remarks to the Author):

The paper has improved and it is nearly ready for publication.

There is still some errors that should be corrected in the new addition of paragraph 4.5 "Errors due to deep water assumption". The paper can not be published without that correction

Equation (7) is not correct for irregular sea states, as there is not a definition of the group velocity in irregular waves.

In a m, kg, s dimensions system, the dimension of E as expressed by equation (8) is:

$$[E] = \text{Kg} \cdot \text{m} / \text{s}^2 = \text{N} \text{ or } \text{J} / \text{m}, \text{ i.e. } E \text{ is energy per unit length}$$

As the dimension of the group velocity is $[c] = \text{m} / \text{s}$, the dimension of P as expressed by equation (7) is:

$$[P] = \text{Kg} \cdot \text{m}^2 / \text{s}^3 = \text{W}$$

i.e. if E is expressed as in equation (8), the expression (7) do not correspond to the energy flux, that should have W/m units.

For equation (7) to be dimensionally correct, E should be the energy per unit of horizontal area $[E] = (\text{J} / \text{m}^2)$.

The energy per unit area of a sea state of scalar spectrum $S(f)$ is given by:

$$E = \rho \cdot g \cdot \int S(f) df = \rho \cdot g \cdot m_0$$

As the zero-moment wave height is defined by:

$$H_{m0} = 4(m_0)^{0.5}$$

them,

$$m_0 = H_{m0}^2 / 16$$

and the energy per unit area is given by:

$$E = \rho \cdot g \cdot H_{m0}^2 / 16$$

If E in equation (8) means the energy per unit area, the wave length L should not appear in the equation.

Please clarify the methodology used to make figure 12 because the paragraph "Using the parameterized mean wavelength, we re-compute the average wave energy flux considering the difference in group wave celerity differentiating between deep, intermediate and shallow water, following the equations in Reeve et al. [2012]." and the wrong equations do not help to understand how results of figure (12) have been obtained.

REVIEWER COMMENTS

Reviewer #1 (Remarks to the Author):

The third draft of the manuscript entitled “Global coastal attenuation of wind-waves observed with radar altimetry” by Passaro et al. Is a clear improvement compared to the initial version, in terms of readability and content.

Just a quick note on some of the rebuttal comments from the authors: although I acknowledge that we have to draw a line at some stage, the fact that some comments on the second draft were not issued during the first review (on the initial draft) does not mean that they should be discarded, as reviewers can also miss some important questions the first time. For a journal such as Nature Communications, articles must be of high quality and therefore all justified comments have to be addressed. The only exception is if comments appear to be contradictory between the two revisions, in which case they are open to debate for clarification.

We appreciate the reviewer’s comment. Indeed, besides mentioning that some comments of reviewer #2 were not issued in the first review, we had nevertheless answered to them as well.

I understand the comments of reviewer #2 on lines 143-144, as the paragraph starting in line 139 with “Statistically significant differences [...]” does not read well: the explanation is a bit muddled, especially with the frequent occurrence of the terms significant and non-significant. An illustration of this is the sentence: “Of the locations not displaying significant difference [...] in the coastal zone”. I strongly suggests that the authors re-read this paragraph and try to make it clearer.

The conclusion of this paragraph lies in the last sentence where it is deduced that the independence of offshore and coastal wave climate corresponds to fetch limited areas. I agree with reviewer #2 that the result sounds too qualitative and uncertain (the words ‘mostly’ and ‘suggesting’ are too vague and not conclusive enough). If this is an important result, then it should be better quantified and backed-up by a more rigorous argumentation, for instance by estimating the fetch at each site and making a statistical analysis with the significant difference. I realise that this could involve a substantial additional work, so I leave it to the authors to decide whether this is feasible (and the best method for it), and/or worth it.

We understand the need to reformulate the content of the paragraph and to avoid vague conclusions. For this reason, we decided to reformulate the quantitative part of the paragraph and leave out the deduction about the areas of coastal amplification of the seasonal cycle constrained to fetch limited areas, since none of these sites show statistical significance of the differences in seasonal amplitude. Therefore, we reformulate the paragraph as follows:

“26% of the locations show a statistically significant (black outline in figure) attenuation of the seasonality. This attenuation is largely consistent with a proportional attenuation of the mean mean SWHs presented in Section 2.1. While there are also areas showing an amplification of seasonality in the coastal sites, the values are not statistically significant.”

Going to main contentious argument, I agree with reviewer #2 that you can get significant errors when calculating the energy flux with mix data source (SWH from altimetry data and wave period from ERA5), and I apology that I overlooked that in my first review. There is a good effort in section 2.3 in justifying the fit between the wave energy fluxes calculated from WAVEWATCH III hindcast and altimetry data. I was just puzzled by the magnitude of the standard deviations in Figure 5b, and

wondered (i) if the mean difference was meaningful in this case, especially near the coast where the energy flux is lowest, and (ii) where these high standard deviations come from.

The mean difference in the last 20 km is indeed so low because both the Wavewatch III hindcast and the altimetry data are able to capture the strong reduction of SWH, which is by an order of magnitude the strongest dependency of the average wave energy flux. The standard deviation observed well matches with the Root Mean Square Differences in SWH found between altimetry and reanalysis values computed at a buoy location in Figure 9 (about 0.20 m). In fact, if we consider the difference in wave energy flux that would be observed for a group velocity of 10 m/s between a mean SWH of 1 m and a mean SWH of 1.2 m, we obtain 2.8 kW/m, which matches the order of magnitude of the standard deviations to which the reviewer is referring in Figure 5b.

A comparison test with energy fluxes calculated purely with ERA5 as suggested by reviewer #2 would have been interesting in theory, but as the authors rightly pointed out, this may not easily be done considering the difference in grid resolution. Another avenue that could be investigated is by estimating the wave period from the backscattering coefficient and significant wave height, following the method proposed by Gommenginger et al. [Gommenginger C.P., Srokosz M.A., Challenor P.G., Cotton P.D. (2003) Measuring ocean wave period with satellite altimeters: a simple empirical model, *Geophys.Res.Letters*, 30, issue 22, DOI: 10.1029/2003GL017743], and successfully validated by Goddijn-Murphy et al. (2015) using Altika altimetry data against wave buoy records. The method was derived in deep water, so I am unsure how applicable this method can be near the coast, but as I said it may be worth investigating it and see how this compares with the values you used.

We do not doubt that in the future the estimation of wave period from satellite altimetry should be considered for further advances. Nevertheless today's state of the art does not have a standard for such derivation.

No mean wave period product from satellite altimetry missions is officially released. The models proposed have different performances and drawbacks (see for example Figure 5 from Badulin et al., 2017). The application of Gommenginger et al., by Goddijn-Murphy et al. (2015) shows an interesting way to go, but it is limited to local characteristics (2 locations, 10 buoys) and concerns an altimeter with different characteristics compared to Jason (i.e. Saral, in Ka band). Even for these locations it was assessed that "The validity of equation (12) and the relation shown in Figure 2 needs to be assessed over intermediate depths between the coast and the ocean" (where equation (12) is the relation used in that study to estimate the wave period).

A similar application for the purpose of our study would require therefore an effort comparable to another full study on the subject. We decided instead to opt for the wave period from ERA5, analyzing and validating the results obtained, which is in-line with that used by Young et al. (2013), who made use of a wavewatch-iii simulation to provide direction and wavenumber (period). We also notice that our main results on the proportional loss in Wave Energy Flux on moving from offshore to coastal is independent of the period T, if we assume that on each offshore-to-coast transit there is one main swell contributing to the Wave Energy Flux.

Therefore, while surely taking the comments into account for future studies based on a more robust estimation of the wave period from altimetry, we also believe that whether we use an altimeter-derived period or a model period in irregular seas will not make a substantial difference to our main results on both the reduction of SWH and of Wave Energy Flux as we near the coast.

In conclusion, I still stand with my previous review where I considered that the subject and the amount of work put in this study justifies being published, but it is also important that the scientific

methods and discussions are very rigorous for the high standards of a scientific journal such as Nature Communications.

There have been some significant improvements from the first draft including a more detailed comparison between model and data calculated energy fluxes, a validation against coastal wave buoys and a discussion on the deep water assumption. The use of the ERA5 period in the energy flux calculations remains questionable but there is no obvious better alternative and the authors made an extensive effort to mitigate it by analysing the model hindcast versus the altimetry data.

After revising the readability of the paragraph starting line 139, as discussed earlier, I would be happy for the manuscript to be published.

We would like to thank the reviewer for the suggestions, the support and the help in improving the manuscript.

Reviewer #3 (Remarks to the Author):

The paper has improved and it is nearly ready for publication.

We would like to thank the reviewer for the suggestions, the support and the help in improving the manuscript.

There is still some errors that should be corrected in the new addition of paragraph 4.5 "Errors due to deep water assumption". The paper can not be published without that correction

Equation (7) is not correct for irregular sea states, as there is not a definition of the group velocity in irregular waves.

In a m, kg, s dimensions system, the dimension of E as expressed by equation (8) is:

$$[E] = \text{Kg} \cdot \text{m} / \text{s}^2 = \text{N} \text{ or } \text{J} / \text{m}, \text{ i.e. } E \text{ is energy per unit length}$$

As the dimension of the group velocity is $[c] = \text{m} / \text{s}$, the dimension of P as expressed by equation (7) is:

$$[P] = \text{Kg} \cdot \text{m}^2 / \text{s}^3 = \text{W}$$

i.e. if E is expressed as in equation (8), the expression (7) do not correspond to the energy flux, that should have W/m units.

For equation (7) to be dimensionally correct, E should be the energy per unit of horizontal area $[E]: (\text{J} / \text{m}^2)$.

The energy per unit area of a sea state of scalar spectrum $S(f)$ is given by:

$$E = \rho_0 \cdot g \cdot \int S(f) df = \rho_0 \cdot g \cdot m_0$$

As the zero-moment wave height is defined by:

$$H_{m0} = 4(m_0)^{0.5}$$

then,

$$m_0 = H_{m0}^2/16$$

and the energy per unit area is given by:

$$E = \rho * g * H_{m0}^2/16$$

If E in equation (8) means the energy per unit area, the wave length L should not appear in the equation.

Please clarify the methodology used to make figure 12 because the paragraph “Using the parameterized mean wavelength, we re-compute the average wave energy flux considering the difference in group wave celerity differentiating between deep, intermediate and shallow water, following the equations in Reeve et al. [2012].” and the wrong equations do not help to understand how results of figure (12) have been obtained.

First of all, the reviewer is of course right and we are thankful for spotting the mistake in the notation of Equation (8). Equation (8) has erroneously the wavelength L inside the formula, which was never applied in our computation. The extract from the MATLAB code (which is available on request as GIT project) is the following:

```
E=rho.*g.* S_J2.swh_1Hz_matrix(cc,zz).^2./16;  
S_J2.mean_wave_power_nodeep(cc,zz)=E.*cg.*10^(-3);
```

Where “rho” is the sea water density, “g” is the gravitational acceleration, “cg” is the group velocity, S_J2.swh_1Hz_matrix(cc,zz) is the significant wave height at a specific point on the matrix containing the along-track location at a specific repeat cycle. The 10⁽⁻³⁾ factor is used to display results in kW/m.

The wavelength is only used to distinguish between deep, intermediate and shallow waters and, as the text says, it is derived using the parametrization of Guo (2002), which we report here:

$$kd = \frac{\sigma^2 d}{g} \left(1 - e^{-\left(\sigma \sqrt{d/g}\right)^{5/2}} \right)^{-2/5}$$

Where sigma = 2 pi tau (tau being the wave period), d is the bathymetry, k is 2pi/lambda and lambda is the wavelength.

The use of the wavelength to distinguish between deep, intermediate (or transitional) and shallow waters is reported in several textbooks including the cited Reeve et al., 2012. In particular, areas in which $d/\lambda \geq 0.5$ are “deep waters”, areas in which $d/\lambda \leq 0.04$ are “shallow waters”, and the areas in between ($d/\lambda < 0.5$ & $d/\lambda > 0.04$) are “transitional waters”.

We clarify this in the text by writing “Using the parameterized mean wavelength L and considering the depth d , we distinguish between deep ($d/L \geq 0.5$), shallow ($d/L \leq 0.04$) and transitional ($0.04 < d/L < 0.5$) waters and compute the corresponding group velocity c_g following Reeve et al. (2012)”

Note that, in order to conform to the most common notation, we now call the group velocity in Equation 7 “ c_g ” instead of “ c ”.

For the benefit of the reviewer, we also report below at the end of these answers the extract from Reeve et al. providing the definition of the different group velocities.

Concerning the comment “Equation (7) is not correct for irregular sea states, as there is not a definition of the group velocity in irregular waves.”,

Equation 7 is the expression that the wave energy flux [W/m] is a product of the integrated energy of the wave spectrum [J/m^2] and the energy propagation velocity (or group velocity [m/s]). In truth, this should be an integral of the product of the frequency dependent group velocity, and the frequency/direction dependent spectral energy. We acknowledge our approach, using integrated wave parameters, is a simplification. However this is a commonly applied simplification, where the spectral distribution of wave energy is not available to estimate the available energy flux.

We follow the approach noted in a recent review by Guillou et al., 2020. <https://doi.org/10.3390/jmse8090705>, as the most common approach taken to estimating wave energy flux, using integrated parameters, maintaining a deep water assumption. Our aim is to assess the order of magnitude of the error/uncertainty introduced into our study, by maintaining a deep-water assumption. The results of this error estimation are of course based on the mean wave field and on integral parameters.

Further to the Guillou et al 2020 review, we refer to Hemer et al., 2017 as a specific example where observations are limited to bulk wave parameters, without spectral information, and estimates of the group velocity are based on a characteristic period and of the depth. The same approach is used in this study.

To clarify this, we add “We notice that a downside of this approach is that, by using Equation (7), we are associating a group velocity to a single characteristic period of an irregular wave field, although the group velocity is a function of a specific frequency of regular waves. This is a commonly applied simplification used where available wave information is limited to bulk wave parameters (such as SWH and mean wave periods) [Guillou et al., 2020]”

Guillou, N., Lavidas, G. and Chapalain, G., 2020. Wave Energy Resource Assessment for Exploitation—A Review. *Journal of Marine Science and Engineering*, 8(9), p.705.

Hemer, M.A., Zieger, S., Durrant, T., O'Grady, J., Hoeke, R.K., McInnes, K.L. and Rosebrock, U., 2017. A revised assessment of Australia's national wave energy resource. *Renewable Energy*, 114, pp.85-107.

[redacted]

See Pg. 53: **Reeve, D., Chadwick, A. J., Fleming, C. (2012). Coastal Engineering: Processes, Theory and Design Practice (2nd ed) E & FN Spon. ISBN9781138060432**

REVIEWERS' COMMENTS

Reviewer #1 (Remarks to the Author):

All comments of the third review have been properly addressed, either by changes in the manuscript, or by comprehensive responses to the questions and suggestions.

I am satisfied for the revised manuscript to be published.

Reviewer #3 (Remarks to the Author):

All reviewer's comments have been taken into account and the paper is nearly ready for publication. I believe that satellite wave data bases will be a very valuable asset for calibration and validation of coastal high resolution wave propagation models that until now should rely on the very sparse coastal buoy data.

Re-revising the last revision, a possible misspelling has been found in page 11, line 178. There referring to figure 5, the following comment is given "Further away, altimetry tends to slightly underestimate the flux,..."

If figure 5b shows the mean difference in average wave energy flux (AWEF) between the altimetry data and reanalysis data, all locations further away of 25 Km show positive values, meaning that the altimetry slightly overestimate the AWEF (or the model data underestimate the AWEF). If this is correct, please make the correction in the manuscript.

REVIEWER COMMENTS

REVIEWERS' COMMENTS

Reviewer #1 (Remarks to the Author):

All comments of the third review have been properly addressed, either by changes in the manuscript, or by comprehensive responses to the questions and suggestions.
I am satisfied for the revised manuscript to be published.

Reviewer #3 (Remarks to the Author):

All reviewer's comments have been taken into account and the paper is nearly ready for publication. I believe that satellite wave data bases will be a very valuable asset for calibration and validation of coastal high resolution wave propagation models that until now should rely on the very sparse coastal buoy data.

Re-revising the last revision, a possible misspelling has been found in page 11, line 178. There referring to figure 5, the following comment is given "Further away, altimetry tends to slightly underestimate the flux,...".

If figure 5b shows the mean difference in average wave energy flux (AWEF) between the altimetry data and reanalysis data, all locations further away of 25 Km show positive values, meaning that the altimetry slightly overestimate the AWEF (or the model data underestimate the AWEF). If this is correct, please make the correction in the manuscript.

Thank you very much for spotting the mistake in the text, we have now changed it into "Further away, altimetry tends to slightly overestimate the flux,..."